# Black carbon radiative effects highly sensitive to emitted particle size when resolving mixing-state diversity

Hitoshi Matsui [1,2], Douglas S. Hamilton [2] & Natalie M. Mahowald [2]

Post-industrial increases in atmospheric black carbon (BC) have a large but uncertain warming contribution to Earth's climate. Particle size and mixing state determine the solar absorption efficiency of BC and also strongly influence how effectively BC is removed, but they have large uncertainties. Here we use a multiple-mixing-state global aerosol microphysics model and show that the sensitivity (range) of present-day BC direct radiative effect, due to current uncertainties in emission size distributions, is amplified 5–7 times (0.18–0.42 W m$^{-2}$) when the diversity in BC mixing state is sufficiently resolved. This amplification is caused by the lifetime, core absorption, and absorption enhancement effects of BC, whose variability is underestimated by 45–70% in a single-mixing-state model representation. We demonstrate that reducing uncertainties in emission size distributions and how they change in the future, while also resolving modeled BC mixing state diversity, is now essential when evaluating BC radiative effects and the effectiveness of BC mitigation on future temperature changes.

[1] Graduate School of Environmental Studies, Nagoya University, Nagoya 464-8601, Japan. [2] Department of Earth and Atmospheric Sciences, Cornell University, Ithaca, NY 14853, USA. Correspondence and requests for materials should be addressed to H.M. (email: matsui@nagoya-u.jp)

**B**lack carbon (BC) aerosol in the atmosphere is estimated to have a large positive climate forcing (heating effect), and its uncertainty is a large hindrance in constraining the human impact on climate change[1–3]. BC particles are gradually coated by non-BC (inorganic and organic) species during atmospheric transport through condensation and coagulation (aging) processes[4,5]. These aging processes enhance BC solar radiation absorption efficiency and the ability of BC to act as cloud condensation nuclei (CCN)[1,6–8]. Enhancements to absorption efficiency and CCN activity then affect BC removal rates and how strongly the surrounding air is heated[1,8–10].

BC particles have a wide range of particle sizes and mixing states (MS), even in the same air mass[4,5], thereby having different absorption efficiency and CCN activity because enhancements to absorption efficiency and CCN activity increase with increasing BC particle size and the fraction of non-BC species[7,9]. Such particle-size- and MS-dependent absorption efficiency and CCN activity are not represented sufficiently in most global aerosol models, which tend to have only 1 or 2 BC MS categories (e.g., hydrophobic, hydrophilic) and 3 or 4 categories for particle sizes[11–17], except for some advanced aerosol models[18,19].

There are large uncertainties in the treatment of aerosol size distributions at emission in current global aerosol models[20,21] and Reddington et al.[20] has previously shown the importance of reducing these uncertainties, especially when estimating aerosol number concentrations—a key parameter in accurately estimating aerosol-radiation and aerosol-cloud interactions. Other previous studies[17,22,23] have also shown the impact of altering assumed emission size distributions on atmospheric BC mass concentrations, but their importance remains unclear. For example, Reddington et al.[22] and He et al.[17] suggested negligible importance of emission size distributions to BC mass, while Bauer et al.[23] showed high sensitivity of BC mass to emission size distributions. In addition, the impact of assumptions about emission size on BC radiative effects is very rarely discussed within the literature.

Here we use a state-of-the-art global aerosol model, which resolves both particle size and MS of BC[10,24,] and show that the direct radiative effect (DRE) of BC is highly sensitive to changes in assumed size distributions at emission[21,25,26], but only when the model resolves BC MS diversity sufficiently. Our simulations show that differences in emission size distributions within their current uncertainty ranges produce a much larger range of BC DRE when the MS variations are sufficiently resolved. The hitherto unrecognized effect occurs because the MS-resolving model simulates three key properties of the BC differently when the MS variations are simulated: atmospheric lifetime of BC, absorption efficiency of BC core, and absorption enhancement by coating species. These results have important implications for evaluating the climate impacts of present-day to future changes in BC and total aerosol DREs.

## Results

### Global aerosol model simulations
We carry out model experiments to quantify the importance of resolving the size-resolved BC mixing state by comparing global simulations using a BC-MS-resolved representation (Multiple-MS) to a single BC MS representation (Single-MS) within the same model framework (see Methods and Supplementary Table 1). In these model experiments all aerosol processes are calculated with the particle-size-resolved and MS-resolved treatment, with Multiple-MS the benchmark simulation that realistically calculates absorption efficiency and CCN activity of BC and their enhancements by aging processes[24,27]. The Single-MS run is designed to represent the simplified BC treatment used in most aerosol models: all

BC-containing particles are treated as thickly-coated BC. With each model set-up, experiments are carried out with the Small, Base, and Large emission size distributions[25,26] (Supplementary Table 2). To put our results in the context of current assumptions in prominent global aerosol models, we also carry out seven additional simulations applying the emission size distributions used by models participating in the Aerosol Comparisons between Observations and Models (AeroCom) project[21]. Only the treatment of emission size distributions for anthropogenic (AN; fossil fuel and biofuel) and biomass burning (BB) sources is different in these ten simulations. The mass flux of aerosol emissions is the same in all simulations (Supplementary Table 3), with the number of particles emitted increasing as the median aerosol size at emission decreases. In the Multiple-MS representation, we assume all BC is emitted as pure BC (Multiple-MS [100%]) to understand the maximum impact of the MS-resolved treatment. We also conduct simulations with two other representations; Multiple-MS [50%], in which we assume half of BC is emitted as pure BC, and Double-MS, which resolves thinly-coated BC and other particles (Supplementary Table 1). Their results are discussed briefly to support our main conclusions obtained by Multiple-MS [100%] and Single-MS.

Global model distributions of BC and other major aerosol species, and their optical properties, have been evaluated against available measurements in previous studies[10]. Our previous studies[27,28] showed that the MS-resolved representation could reproduce important observed features of BC MS, such as number fractions of BC-containing and BC-free particles and coating amounts of BC-containing particles with their size dependences, which are neglected in the Single-MS representation. Unless noted otherwise, global and 5-year mean values are shown in this study.

### Sensitivity of BC DRE
The Small-size simulations have a longer BC lifetime and higher BC burden than the Large-size simulations for both the Multiple-MS and Single-MS representations because larger particles generally have higher CCN activity (lower critical supersaturation required to form a cloud droplet), which translates into a shorter BC lifetime since wet removal is the dominant BC sink[24,29]. Simulated BC lifetimes in both Multiple-MS (4.5–6.0 d) and Single-MS (4.3–5.1 d) tend to be shorter than the AeroCom multi-model average (6.5 d)[30], which is generally consistent with previous studies[30,31], suggesting a too long BC lifetime in many AeroCom models. However, the overestimation of upper tropospheric BC in many global aerosol models[30,32] is also seen in the Multiple-MS and Single-MS simulations shown here (Supplementary Fig. 1).

To examine the sensitivity of BC burden and its radiative effect to assumed size distributions in emissions, we use the ratio of global-mean BC values between the Small-size and Large-size simulations (Small/Large ratio). The BC lifetime in the Multiple-MS representation is slightly more sensitive to the uncertainty in emission size distributions (Small/Large ratio of 1.32) than that in the Single-MS representation (Small/Large ratio of 1.19) (Fig. 1a). This stronger BC lifetime sensitivity results from Multiple-MS-resolving pure BC and thinly-coated BC, which have larger variability in CCN activity than thickly-coated BC within the range of particle sizes in the Small-size and Large-size simulations[9]. This tendency of BC lifetime to be longer and BC burden to be larger with smaller emission size is qualitatively consistent with Bauer et al.[23].

BC and organic aerosol are sensitive to emission size distributions (for AN and BB sources), whereas other aerosol species (sulfate, nitrate, dust, and sea salt) and the aerosol optical depth appear insensitive to emission size distributions

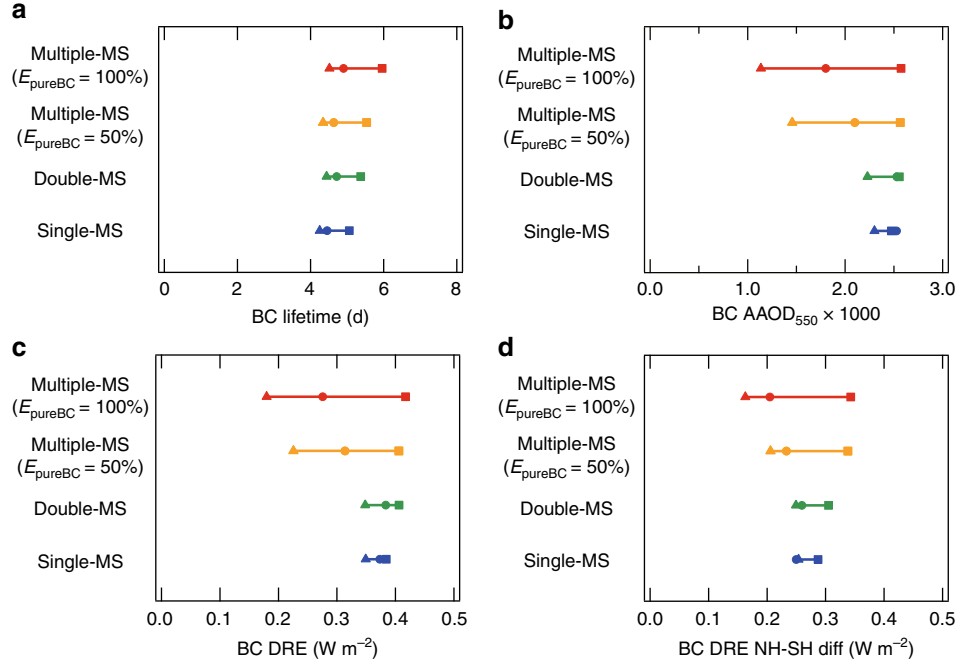

**Fig. 1** Sensitivity of black carbon properties to emission size distributions. The maximum-minimum ranges for **a** black carbon (BC) lifetime, **b** BC absorption aerosol optical depth at 550 nm (AAOD$_{550}$), **c** BC direct radiative effect (DRE), and **d** the contrast of BC DRE between the Northern and Southern Hemispheres. Squares, circles, and triangles show the values in the Small-size, Base-size, and Large-size simulations, respectively

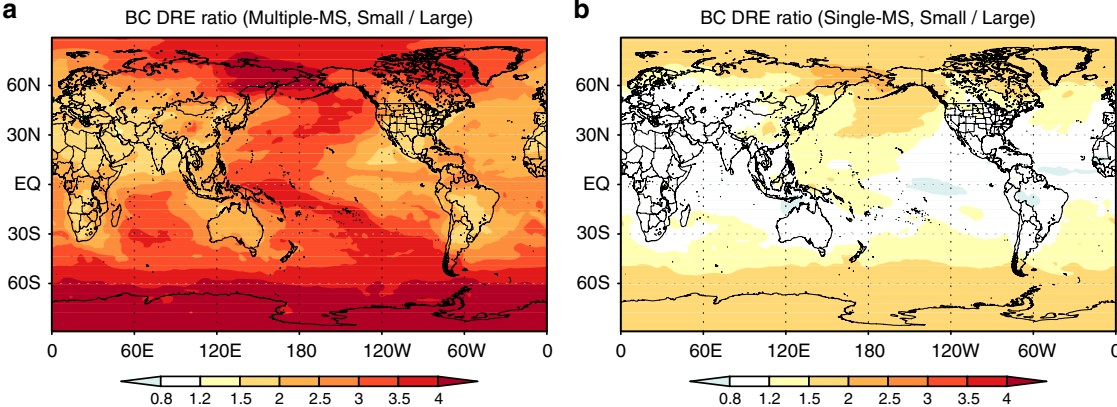

**Fig. 2** Spatial distributions of the sensitivity of black carbon direct radiative effect. The ratio (dimensionless) of black carbon (BC) direct radiative effect (DRE) between the Small-size and Large-size simulations for the **a** Multiple-mixing-state and **b** Single-mixing-state representations

(Supplementary Figs. 2a, b). While total CCN number concentrations are sensitive to assumptions about emission size distributions[20], the Multiple-MS and Single-MS representations exhibit similar sensitivity in total CCN number concentrations as BC-free or thickly-coated BC particles dominate the aerosol population (Supplementary Fig. 2d, e).

Mass absorption cross section (MAC) has large spatial and temporal variations: typically between 7 and 13 m$^2$ g$^{-1}$ at 550 nm in measurements and an average of ~8 m$^2$ g$^{-1}$ at 550 nm in AeroCom models[1,33–37]. Global-mean MAC is 8.8 and 13.6 m$^2$ g$^{-1}$ in Multiple-MS and Single-MS (Base-size simulations), respectively (Supplementary Table 3), showing that MAC in Multiple-MS is within the typical observed range, while MAC in Single-MS is above the typical observed range.

BC absorption aerosol optical depth at 550 nm (AAOD$_{550}$) and DRE in Multiple-MS (0.0018 and 0.28 W m$^{-2}$ in the Base-size

simulation) are 25% lower than those in Single-MS (0.0024 and 0.36 W m$^{-2}$), because the enhancement of absorption by coating species is overestimated in Single-MS[10] (Fig. 1b, c). The Small/Large ratio of both AAOD$_{550}$ and DRE doubles from 1.1 in Single-MS to 2.3 in Multiple-MS, reflecting the higher sensitivity of AAOD$_{550}$ and DRE to emission size distributions in Multiple-MS than Single-MS.

The Small/Large ratio of DRE in Multiple-MS is higher than that in Single-MS all over the globe (Fig. 2 and Supplementary Figs. 3 and 4). Multiple-MS and Single-MS have similar spatial patterns: Small/Large ratio is lower near source regions at low and mid-latitudes and higher over remote regions and at high-latitudes (Fig. 2). These results show the sensitivity of BC DRE to emission size distributions is enhanced by the aging processes during transport, especially when MS is sufficiently resolved. The contrast of BC DRE between the Northern and Southern

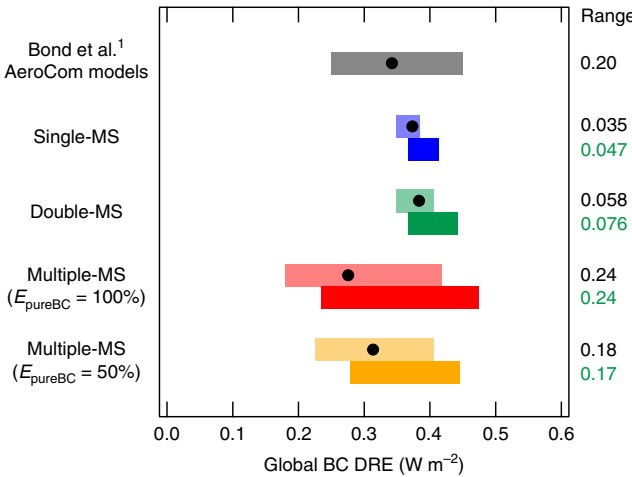

**Fig. 3** Sensitivity of black carbon direct radiative effect to emission size distributions. Black carbon (BC) direct radiative effect (DRE) (circles, Base-size simulations) and its ranges (horizontal bars). The ranges are shown for the Small-size and Large-size simulations (upper bars) and the simulations using the emission size distributions of the AeroCom models (lower bars). The mean BC DRE and its inter-model variability by the AeroCom model simulations[1] are shown for comparison (gray bar)

Hemispheres is a factor of 2 difference between the Small-size and Large-size simulations in Multiple-MS (Fig. 1d). In Single-MS, this contrast is not sensitive to emission size distributions. A multiple MS-resolved treatment is therefore important for accurate calculations of the contrast of BC DRE between the Northern and Southern Hemispheres and between land and ocean (Figs. 1d and 2 and Supplementary Fig. 4).

The BC DRE ranges estimated from the Small-size and Large-size simulations and when incorporating the range of emission size distributions used in the AeroCom models are both 0.24 W m$^{-2}$ in the simulations with the Multiple-MS representation (Fig. 3 and Supplementary Fig. 5). These DRE ranges are slightly larger than the multi-model variability in BC DRE simulated by the AeroCom models[1] (0.20 W m$^{-2}$), which contains many sources of uncertainty, including, but not limited to, differences in structural, process, chemical, and physical parameterizations (see Methods). These ranges for Multiple-MS, which are not recognized currently, are 5–7 times greater than those for Single-MS (Fig. 3). Therefore, many global aerosol models, which use an aerosol representation similar to Single-MS, will significantly underestimate the magnitude of the BC DRE uncertainty.

The simulations in which we assume half of BC is emitted as pure BC (Multiple-MS [50%]) also have a high sensitivity of AAOD$_{550}$ and DRE to emission size distributions (yellow lines in Figs. 1 and 3), though the simulations have a 20% lower sensitivity than the Multiple-MS [100%] simulations. Global distributions of AAOD$_{550}$ and DRE are also similar between Multiple-MS [100%] and [50%] (Supplementary Figs 6 and 7). These results show that our main conclusions are insensitive to the exact BC mixing state assumed at emission.

The simulations assuming two MS categories (Double-MS; thinly-coated BC and others) (Supplementary Table 1) have a slightly higher sensitivity of BC lifetime, AAOD$_{550}$, and DRE to emission size distributions than the Single-MS simulations (green lines in Figs. 1 and 3) because Double-MS resolves thinly-coated BC, which has a longer lifetime and lower MAC than the internally-mixed BC represented in Single-MS. However, as shown in Figs. 1 and 3, the sensitivity of Double-MS and Single-MS is similar for all BC properties. This result suggests that most

global aerosol models, which use a BC representation similar to Double-MS or Single-MS, currently underestimate the sensitivity of BC properties with respect to emission size distributions.

**Causes of high sensitivity by mixing state diversity.** To explain why Single-MS is under-representing the BC DRE range, we decompose BC AAOD$_{550}$ into three effects: (1) the BC lifetime effect, (2) absorption efficiency of BC core (mass absorption cross section of BC core (MAC$_{core}$)), and (3) absorption enhancement by coating species ($E_{abs,coat}$).

$$\overline{AAOD}_{550,core+coat} = \bar{B} \times \overline{MAC}_{core} \times \bar{E}_{abs,coat} \qquad (1)$$

$$\overline{MAC}_{core} = \frac{\overline{AAOD}_{550,core}}{\bar{B}} \qquad (2)$$

$$\bar{E}_{abs,coat} = \frac{\overline{AAOD}_{550,core+coat}}{\overline{AAOD}_{550,core}} \qquad (3)$$

where AAOD$_{550,core+coat}$ and AAOD$_{550,core}$ are BC AAOD$_{550}$ with and without coating species, respectively, and $B$ is BC burden (g m$^{-2}$). BC burden is inversely related to BC lifetime, since the mass flux of emissions is the same in all simulations. We compare the sensitivity of each effect between the Multiple-MS and Single-MS representations (Fig. 4).

First, BC lifetime in Multiple-MS is more sensitive to emission size distributions, as shown above (Fig. 1a). The variability range of BC lifetime is underestimated by 44% in the Single-MS representation (4.5–6.0 d for Multiple-MS and 4.3–5.1 d for Single-MS; Fig. 1a) because explicitly calculating the size-dependent and MS-dependent critical supersaturation of BC-containing particles in the Multiple-MS covers a broader range of CCN activity (including the less CCN-active population of particles such as pure BC and thinly-coated BC). In Single-MS only a size-dependent critical supersaturation is calculated so all BC-containing particles are more CCN-active than the pure BC and thinly-coated BC resolved in Multiple-MS. The lifetime effect explains a part of the different sensitivity of AAOD$_{550}$ and DRE between Multiple-MS and Single-MS (Fig. 1b, c; Eq. (1)).

Second, the mass absorption cross section of BC core (MAC$_{core}$) in Multiple-MS is more sensitive to emission size distributions: Small/Large ratio is 1.41 for Multiple-MS and 0.95 for Single-MS. The range of MAC$_{core}$ is underestimated by 72% in the Single-MS representation (4.2–5.9 m$^2$ g$^{-1}$ for Multiple-MS and 5.4–5.9 m$^2$ g$^{-1}$ for Single-MS (Fig. 5a)). The mass median diameter of BC core (see Methods) in Multiple-MS is ~200 nm (Base-size simulation) (Fig. 5a), consistent with particle-resolved measurements[4,5,38]. In contrast, the mass median diameter of BC core in Single-MS is ~90 nm. The number fraction of BC-free particles to the total aerosol population is 70–80% in East Asia, 75–90% in Europe, and 90% over the Arctic in both measurements[22,27,39] and Multiple-MS, whereas BC-free particles are not treated (0%) in Single-MS unless BC is negligible. The inclusion of BC-free particles in Multiple-MS leads to a larger and more realistic BC amount (diameter) in each BC particle (Fig. 4; middle box). The treatment of BC-free particles in Multiple-MS enhances the sensitivity of BC core size and MAC$_{core}$ to emission size distributions by 3–4 times, compared with Single-MS (vertical bars in Fig. 5a). In Multiple-MS the Small-size simulation has larger MAC$_{core}$, following the Mie theory curve (black line in Fig. 5a), and this response amplifies the DRE range caused by the lifetime effect. In contrast, in the Single-MS simulations, the response of MAC$_{core}$ is opposite and partially cancels the DRE range caused by the lifetime effect.

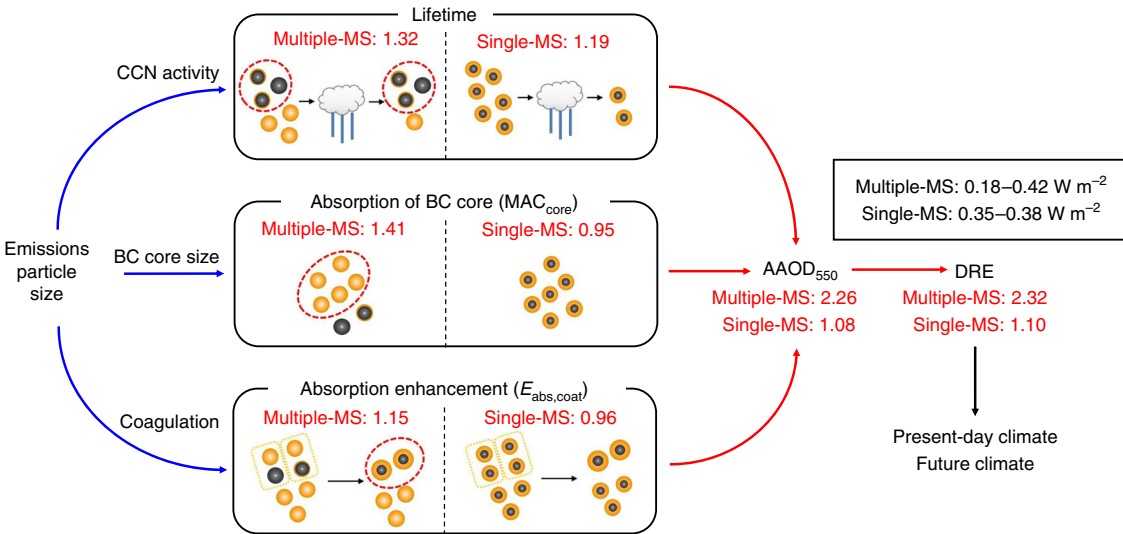

**Fig. 4** Causes of high sensitivity of black carbon direct radiative effect by resolving mixing state. The flowchart shows why the Multiple-mixing-state representation results in high sensitivity of black carbon (BC) direct radiative effect (DRE) due to emission size distributions. Blue and red arrows show negative and positive responses, respectively, in the Multiple-mixing-state representation. Specifically, decreasing particle sizes at emission (e.g., Small-size simulation) increases BC lifetime, BC core absorption, and absorption enhancement, and these enhancements increase absorption aerosol optical depth at 550 nm ($AAOD_{550}$) and DRE of BC in the Multiple-mixing-state representation. The values shown by red are the ratios between the Small-size and Large-size simulations. The representation of particles in red circles is especially important for accurate calculations of the lifetime (pure BC and thinly-coated BC particles), BC core absorption (BC-free particles), and absorption enhancement (thickly-coated BC particles) effects, which are the causes of different BC DRE sensitivity between the Multiple-mixing-state and Single-mixing-state representations

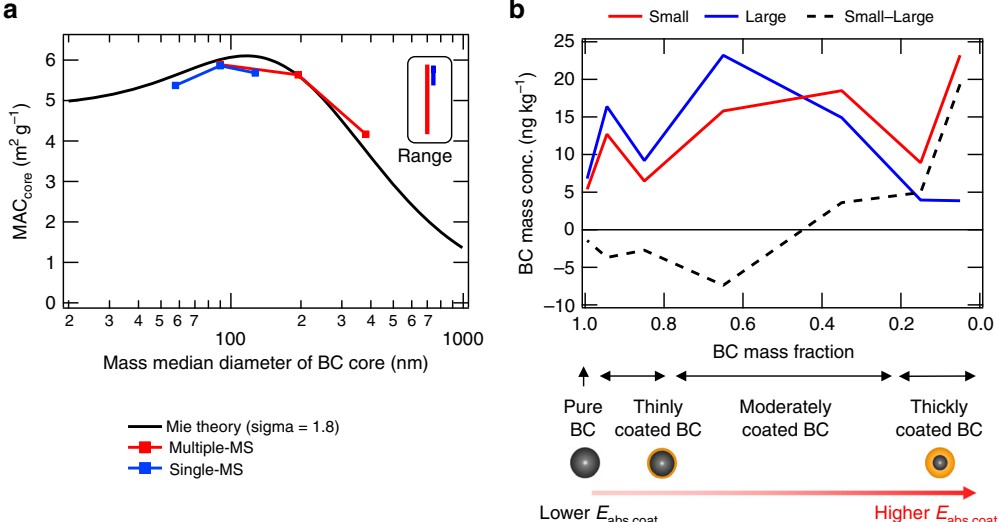

**Fig. 5** Key variables for the high sensitivity in black carbon direct radiative effect. **a** Size dependence of the mass absorption cross section of black carbon (BC) core ($MAC_{core}$, Eq. (2)) for the Multiple-mixing-state and Single-mixing-state representations. Squares correspond to the Small-size, Base-size, and Large-size simulations (at the surface). Black line shows theoretical calculations (using the algorithms of Bohren and Huffman[68] based on the Mie theory) assuming lognormal size distributions (sigma = 1.8). The ranges of $MAC_{core}$ are shown by red (Multiple-mixing-state) and blue (Single-mixing-state) vertical bars. **b** BC mixing state distributions of BC mass for the Small-size (red) and Large-size (blue) simulations in the Multiple-mixing-state representation (at the surface). BC particles are gradually shifted to the right by aging processes in the atmosphere

Third, the absorption enhancement due to the coating species ($E_{abs,coat}$) in Multiple-MS is more sensitive to emission size distributions: Small/Large ratio is 1.15 for Multiple-MS and 0.96 for Single-MS. The variability range of $E_{abs,coat}$ is underestimated by 63% in the Single-MS representation (1.5–1.8 for Multiple-MS and 2.2–2.3 for Single-MS; Supplementary Table 3). The Small-size simulation in Multiple-MS has ~5 times more thickly-coated BC particles (which have a low BC mass fraction) than the Large-size simulation (Fig. 5b). These thickly-coated BC particles are mainly produced by coagulation[22], which is faster in the Small-size simulation compared to the Large-size simulation because total aerosol number concentrations are higher (i.e., more smaller particles) and hence the rate of coagulation is faster. Since thickly-coated BC particles have a higher absorption enhancement compared to thinly-coated BC particles, absorption enhancement in the Small-size simulation

is higher than that in the Large-size simulation. Global mean $E_{abs,coat}$ is 1.8 and 1.5 in the Small-size and Large-size simulations, respectively. Global mean $E_{abs,coat}$ in the Base-size simulation (1.6) is similar to that in a recent global model simulation (1.5) which used the results of a particle-resolved box model[40]. Single-MS has higher $E_{abs,coat}$ (2.2–2.3) and cannot obtain the difference due to thickly-coated BC concentrations between the Small-size and Large-size simulations because all BC-containing particles are treated as thickly-coated BC.

The core-shell MS treatment (BC makes the core while other species make the shell) is used for internally-mixed BC particles in our simulations[24]. This treatment is likely to be realistic because limited particle-resolved measurements have shown that attached-type BC particles (BC is at the surface of the particle), which have lower enhancement of absorption than coated-type BC particles (BC is inside of the particle)[41], are very limited (<3%) in the atmosphere[42,43]. The conclusions obtained in this study do not change even if we use different MS treatments for internally-mixed BC particles (see Methods and Supplementary Fig. 8). Some field measurements have reported low $E_{abs,coat}$ values[44,45], however they are not able to observe absorption enhancement in humid regions of the atmosphere as ambient particles are usually dried (relative humidity of < 50%) in field measurements[40]. Some laboratory experiments[46] and global modeling studies[40,47] have shown the importance of BC absorption being enhanced by the presence of water in the particle.

In the Multiple-MS simulations, all three effects (lifetime, core absorption, and absorption enhancement) increase absorption with decreasing emission particle size, and these responses amplify the range of BC $AAOD_{550}$ and DRE (Fig. 4). The Small/Large ratio of $AAOD_{550}$ and DRE can be explained by the product of the Small/Large ratios for the three effects for both Multiple-MS and Single-MS (Fig. 4). Core absorption and absorption enhancement effects are the main causes of the 5–7 times higher sensitivity of BC DRE in Multiple-MS (than Single-MS).

To show the importance of resolving BC-free particles, we conducted offline optical calculations, assuming 2 and 3 MS categories, using the results of Multiple-MS simulations. In the 2 MS treatment, we consider hydrophobic and hydrophilic BC groups (similar to Double-MS, BC mass fractions are assumed as 0.9–1.0 and 0–0.9, respectively) from the 8 MS information in Multiple-MS. In the 3-MS treatment, hydrophobic BC, hydrophilic BC, and BC-free groups (BC mass fractions are assumed as 0.9–1.0, 0.0001–0.9, and 0–0.0001, respectively) are considered. The Small/Large ratios in the 2 MS treatment are 1.04 and 0.94 for the BC core absorption and absorption enhancement effects, respectively, suggesting that the 2 MS treatment cannot capture the sensitivity of these two effects as simulated by Multiple-MS (Small/Large ratios are 1.41 and 1.15, respectively, in Multiple-MS). The Small/Large ratios in the 3 MS treatment are 1.24 and 1.19 for the two effects, showing a much increased sensitivity for the two effects by the addition of resolving BC-free particles. However, the sensitivity for the BC core absorption effect is still underestimated (by 41%) compared with the Multiple-MS treatment. An important component of most global aerosol models with 2 BC MS categories is the chosen threshold value to distinguish between hydrophobic and hydrophilic BC. This aging threshold term is highly variable, and often modified in order to better correlate simulated BC properties with observations. Therefore, resolving multiple BC MS categories removes the dependence of choosing a threshold and represents the diversity of CCN activity and absorption enhancement of BC-containing particles more realistically.

**Sensitivity to emission changes in the future**. Global BC emissions are estimated to decrease for AN sources[48] and increase for BB sources in the future[49,50]. We show how BC and total DREs respond to these potential future emission changes. In Multiple-MS, the range of BC DRE efficiency (defined as the change in BC DRE per unit BC emission change, see Methods) is 2.7 and 2.8 times larger than Single-MS for AN and BB sources, respectively, within the uncertainty range of emission size distributions (Fig. 6a). Total DRE (scaled to multi-model estimates of direct radiative forcing in Myhre et al.[3], see Methods) in Multiple-MS is 17 and 2.6 times more sensitive to the changes in AN and BB emissions, respectively, compared with Single-MS (Fig. 6c).

In the Base-size simulations, BC DRE efficiency for AN sources in Multiple-MS is smaller than that in Single-MS (Fig. 6a). This suggests that model predictions of how future reductions to AN BC emissions will reduce BC warming are likely to be overestimated if BC MS is not sufficiently resolved. In Multiple-MS, total DRE efficiency for BB sources is positive in the Small-size simulation, whereas it is negative in the Large-size simulation, suggesting that the treatment of emission size distributions will be important to determine the sign (warming or cooling effect) of total DRE for BB sources in the future climate (Fig. 6c). Total DRE for BB sources changes from positive (Single-MS) to negative (Multiple-MS) by resolving BC MS (in Base-size simulations). Though BB emissions also change aerosol-cloud interactions and rapid adjustments, this result suggests that the enhancement of BB emissions in the future will have an overall cooling impact on climate.

We assumed no changes in emission size distributions between the present-day and future (reduced or enhanced emissions) simulations in Fig. 6a, c. When emission fluxes are changed with changing emission size distributions (for all sources) from the Base-size (present-day) to the Large-size (future), both BC and total aerosol DREs have negative values for both AN and BB sources (triangles in Fig. 6b, d). The negative BC DRE is calculated (despite an increase in BB BC emissions) because the reduction of positive AN BC DRE, by changing emission sizes from the Base-size to the Large-size, exceeds the enhancement of positive BB BC DRE. In contrast, when emission fluxes are changed with changing emission size distributions from the Base-size to the Small-size, both BC and total aerosol DREs have positive values for both AN and BB sources (squares in Fig. 6b, d). The positive BC DRE is calculated (despite a decrease in AN BC emissions) because the enhancement of BB BC DRE by changing emission sizes from the Base-size to the Small-size is comparable to the reduction of AN BC DRE (by the reduction of emission fluxes). Our results show that the present-day to future changes of BC and total aerosol DREs are 6–13 and 4 times more sensitive to changes in emission size distributions, respectively, by resolving MS diversity with a potential to change the sign of their DREs (warming or cooling effect) for both AN and BB sources.

**Discussion**

Our results suggest current aerosol models which are frequently used in multi-model inter-comparison studies underestimate the multi-model range of BC DRE because most of them do not resolve BC MS sufficiently. A more realistic representation of MS which includes pure BC, thinly-coated BC, thickly-coated BC, and BC-free particles, is essential for accurate calculations of the lifetime, core absorption, and absorption enhancement effects and BC DRE. Our results show that altering microphysical processes changes BC DRE significantly even when similar total (bulk) mass concentrations of BC and non-BC are simulated.

The interactions of BC with atmospheric large-scale circulation and precipitation pattern such as the Hadley circulation, the

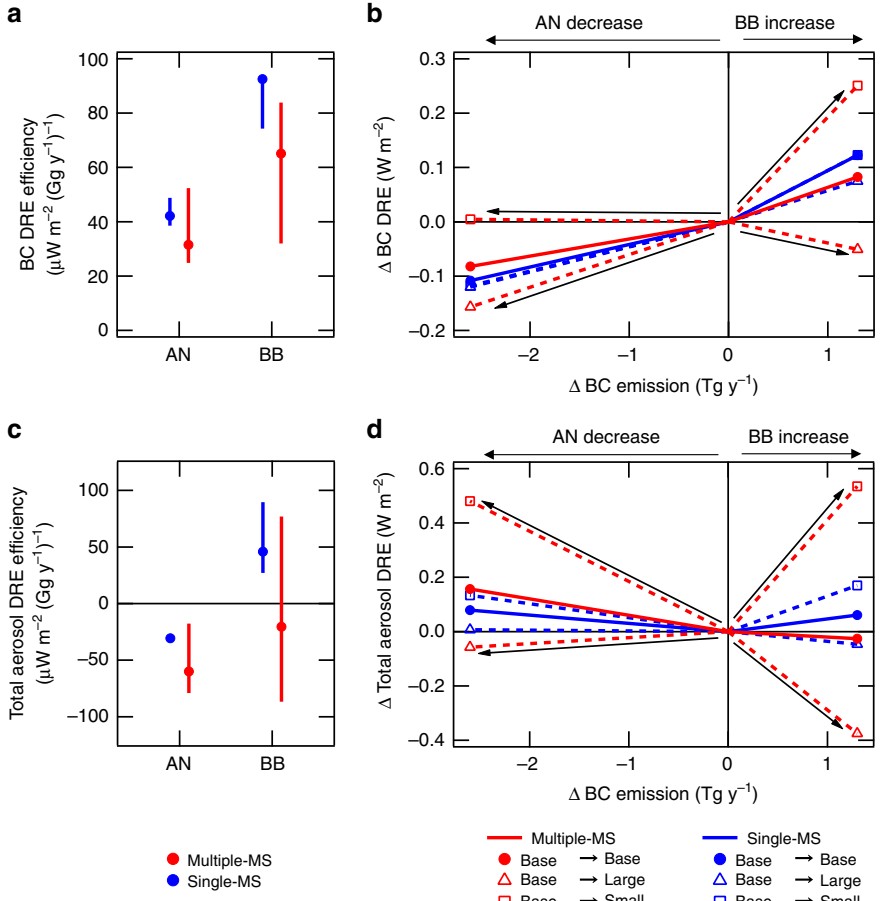

**Fig. 6** Sensitivity of direct radiative effect to emission changes in the future. **a, c** Direct radiative effect (DRE) efficiency (see Methods) is shown for (**a**) black carbon (BC) and (**c**) total aerosols (sum of BC, organic aerosol, sulfate, and nitrate) for anthropogenic (AN) and biomass burning (BB) sources. Vertical bars show the maximum-minimum ranges of the Base-size, Small-size, and Large-size simulations. The values for the Base-size simulations are shown by circles. **b, d** Responses of (**b**) BC and (**d**) total aerosol DRE to changes in AN and BB emission fluxes. Changes in DRE when emission size distributions (for all sources) are changed from the Base-size (present-day) to the Base-size, Small-size, and Large-size (future) are shown by closed circles, open squares, and open triangles, respectively. Total aerosol DRE and its efficiency in (**c**) and (**d**) are calculated with the scaling of BC and non-BC radiative forcing to radiative forcings in Myhre et al.[3] (see Methods)

Intertropical Convergence Zone (ITCZ) and monsoon systems may be more sensitive to aerosol size distributions and MS than currently thought (Figs. 1d and 2). For example, a larger BC DRE may shift the ITCZ to the north through significant changes in surface temperature, meridional sensible and latent heat transport, and convective fluxes[51,52]. The interactions of BC with ice and snow surface at high-latitudes may also be more sensitive to aerosol size distributions and MS than currently thought (Fig. 2 and Supplementary Fig. 3)[53,54]. A reevaluation of the role of BC in the Earth system using aerosol models which can resolve many MS categories is therefore needed.

The fractions of BC to co-emitted aerosol and precursor gas species (e.g., sulfur dioxide, organic aerosol) in future emission reductions are important in evaluating the effectiveness of BC mitigation[1,55,56]. In addition to these fractions, our results show that emission size distributions and their interactions with MS diversity (through microphysical processes and optical properties) are essential components needed in order to improve confidence in the effectiveness of future BC mitigation strategies. A larger reduction of biofuel emissions compared to fossil fuel emissions[48] will result in total AN emissions shifting to smaller size distributions in the future, as biofuel emissions are generally larger in size compared with fossil

fuel emissions[21,25,26], likely resulting in a warming effect (more positive DRE). The size distribution of BB emissions may also change in the future due to changes in fire regimes; for example, smoldering-phase fires have a smaller aerosol size distribution than flaming-phase fires[57,58]. Our MS-resolved simulations show that reducing smaller particles from emissions preferentially is likely an effective way to induce a larger cooling effect (increased negative DRE) on climate by future aerosol reductions (Fig. 6b, d).

Recent studies have suggested that the rapid adjustment (aerosol induced cloud changes) associated with BC may be important for explaining a low temperature response from BC[59–61]. Since the global mean temperature change from BC in global climate models is lower than expected[62], the rapid adjustment may be an important effect required to obtain the full picture of how BC modifies climate and hence understanding future mitigation policy impacts. The estimation of the rapid adjustment of BC is much more uncertain than that of BC DRE. Here we suggest that 44 ± 8%, 32 ± 6%, and 25 ± 10% of BC DRE may be offset by the rapid adjustment in the Small-size, Base-size, and Large-size simulations (Methods, Supplementary Fig. 9, and Supplementary Table 4), noting that the offset percentage increases with decreasing emission sizes

(Supplementary Table 4). If we assume these percentages can be applied to our BC DRE estimates, the Small/Large ratios of BC effective radiative forcing (BC DRE + rapid adjustment) are reduced to 1.7 and 0.80 for Multiple-MS and Single-MS, respectively (2.3 and 1.1, respectively, for BC DRE only). These results suggest accurate estimates of the rapid adjustment combined with BC MS diversity will be important in future studies to constrain BC climate impact.

Contrary to the current understanding, our findings show that BC and total aerosol DREs and their future changes cannot be calculated accurately without the following two points: (1) reducing large uncertainties in aerosol size distributions in emissions by more observation data and (2) representing interactions of aerosol size distributions with BC MS; both of which have been omitted or largely oversimplified in most global climate and aerosol models. In addition to reducing other uncertainties of BC such as emission mass flux, aerosol-cloud interactions, and rapid adjustments, a more detailed representation of aerosol size distributions and MS is therefore required when evaluating the effectiveness of BC mitigation in order to make effective climate change policy; otherwise an oversimplified representation of BC MS will lead to incorrect estimates of BC DRE impacts, reducing confidence in both the magnitude and sign of resulting future temperature changes when evaluating BC climate mitigation strategies.

## Methods

**Global model simulations (online simulations).** We used the Community Atmosphere Model version 5 (CAM5)[63,64] with the Aerosol Two-dimensional bin module for foRmation and Aging Simulation version 2 (CAM5-chem/ATRAS2)[10,24]. Online simulations were conducted for 6 years by using the stand-alone version of CAM5 (FC5 compset) in the Community Earth System Model (CESM) version 1.2.0 with present day climatological data (1982–2001) for sea surface temperature and sea ice. The horizontal resolution is $1.9° \times 2.5°$ with 30 vertical layers from the surface to ~40 km. The last 5 years were used for analysis. Emission data (mass flux) for aerosol and gas-phase species follow Matsui and Mahowald:[10] AN, BB, and biogenic emissions for the year 2000[65] and online emissions of dust and sea salt. The CAM5-chem/ATRAS2 model considers emissions, gas-phase chemistry, condensation and evaporation of both inorganic and organic species, coagulation, nucleation, activation of aerosol and evaporation from cloud, aerosol formation in clouds, dry and wet deposition, aerosol optical properties, aerosol-radiation interactions, and aerosol-cloud interactions. Aerosol optical properties are calculated by the core-shell treatment for internally-mixed BC particles and the well-mixed treatment for pure BC and BC-free particles[24]. Mass concentrations of eight aerosol species (sulfate, nitrate, ammonium, dust, sea salt, primary and secondary organic aerosol, BC, water) and number concentrations are calculated in the model.

This study mainly used two BC MS representations: (1) Multiple-MS and (2) Single-MS (Supplementary Table 1) which correspond to ATRAS-d and ATRAS-b, respectively, in Matsui and Mahowald[10]. In Multiple-MS, aerosol particles are resolved with a two-dimensional (2-D) bin representation: 12 size bins from 1 to 10,000 nm in aerosol dry diameter and 8 BC MS bins. BC MS bins are defined by the fraction of BC to total dry aerosol mass for fine particles (between 40 and 1250 nm in dry diameter): pure BC particles have a mass fraction of 0.99–1.00, BC-free particles have a mass fraction of 0.0–0.0001, and six categories for internally-mixed BC particles have mass fractions of 0.0001–0.10, 0.10–0.20, 0.20–0.50, 0.50–0.80, 0.80–0.90, and 0.90–0.99. Multiple-MS can calculate the shift of particles within the 2-D bins by condensation and coagulation processes explicitly. Since all aerosol processes (e.g., CCN activity, optical property, activation, deposition) are calculated using the information of the 2-D bins, Multiple-MS can calculate absorption efficiency and CCN activity of BC and their enhancements by aging processes in detail by resolving the diversity of BC particles in the atmosphere. In Single-MS, aerosol particles are resolved with a one-dimensional bin representation: 12 size bins (from 1 to 10,000 nm in dry diameter) with a single BC MS. All particles are assumed to be internally-mixed BC in this representation. Therefore, Single-MS cannot resolve the diversity of BC particles sufficiently. In addition to Multiple-MS and Single-MS, we also used Double-MS in which thinly-coated BC particles (BC mass fraction of 0.90–1.00) and the other particles are resolved (Supplementary Table 1). Since most global aerosol models use 1 or 2 BC MS categories[11–17], except for some advanced models[18,19], the aerosol representation of most global aerosol models is similar to Single-MS or Double-MS.

The treatment of size distributions in emissions has large uncertainties in current global aerosol models[21]. We defined their uncertainties for fossil fuel, biofuel, and BB sources based on previous studies[21,25,26]. We conducted the

Small-size, Base-size, and Large-size simulations using the uncertainty range reported in Lee et al.[25] and Carslaw et al.[26] (Supplementary Table 2). These simulations were used to interpret the response and sensitivity of BC variables (lifetime, AAOD_550, and DRE) to changes in emission size distributions. Seven additional simulations were conducted based on emission size distributions used by AeroCom models[21] (Supplementary Table 2). These simulations were used to support the conclusions obtained from the Small-size, Base-size, and Large-size simulations and to understand how the BC DRE estimation was influenced by the variability range of emission size distributions used in current global aerosol models. We conducted ten simulations for each MS representation (Supplementary Table 2). Only the treatment of emission size distributions for fossil fuel, biofuel, and BB sources is different in these ten simulations. Other settings (including emission mass flux) are the same for these simulations.

As described in Lee et al.[25], carbonaceous aerosol particle size is a very uncertain parameter because the size of combustion particles from fossil fuels depends on the details of the source (e.g., fuel and industry type and age of the facility) and the size from BB and wildfires depends on the burning efficiency of the fuel and its properties (e.g., moisture content and fuel type). This uncertainty range for these emissions particle sizes corresponds to the maximum-minimum ranges of median diameters of emissions assumed in the AeroCom models (Supplementary Table 2). The range of median diameters for fossil fuel sources is 30–80 nm in both Lee et al.[25] and the AeroCom models. The emissions particle size range for biofuel and BB sources in Lee et al.[25] (50–200 nm) is at slightly larger sizes compared to the diversity among the AeroCom models (30–150 nm), and a more recent aerosol forcing uncertainty study (by the same group) suggests emitted carbonaceous particles from BB may be as large as 300 nm diameter[66].

We used two different treatments of Multiple-MS considering the uncertainty in BC MS in emissions (Supplementary Table 1): (1) simulations assuming all BC particles are emitted as pure BC (mainly used in the main text) and (2) simulations assuming 50% of BC is emitted as pure BC and the other 50% of BC is emitted as internally-mixed BC. In the simulations assuming 50% of BC is emitted as internally-mixed BC, the shell (total) to core (BC) diameter ratio of internally-mixed BC particles in emissions was assumed to be 1.1 for fossil fuel sources and 1.4 for biofuel and BB sources based on particle-resolved BC measurements[27,58,67]; these values have large uncertainty currently.

**Definition of BC variables.** BC AAOD_550 (5-year mean) was calculated in both online simulations and offline calculations (shown later). BC AAOD_550 was defined as the difference of AAOD_550 between when BC was considered and when BC was excluded.

BC DRE (5-year mean) was calculated in online simulations only. It was defined based on Ghan[68] (for all sky condition) as the difference of shortwave radiative flux at the top of atmosphere between when BC was considered (base radiative transfer calculations) and when BC was excluded (diagnostic radiative transfer calculations).

Mass absorption cross section (MAC) was defined as the ratio of BC AAOD_550 to column BC (5-year mean). Global-mean MAC was calculated from the ratio of global-mean BC AAOD_550 to global-mean column BC. In Fig. 5a, we used MAC for BC core (MAC_core) which was calculated from BC AAOD_550 with BC core only (obtained by offline calculations).

BC core size at the surface was calculated from monthly outputs of online simulations. It was calculated diagnostically from volume concentrations of BC (mass/density) and number concentrations for each horizontal grid (at the surface) and each size and MS bin. The mass median diameter of BC core size (used in Fig. 5a) was calculated as BC core size at median BC mass from all data (N of horizontal grid cells × N of aerosol bins × N of months). This means half of BC mass has larger (or smaller) BC core diameters than the mass median diameter.

The Small/Large ratio was defined as the ratio of 5-year mean values between the Small-size and Large-size simulations. The range was defined as the difference between maximum and minimum values (5-year mean) for both the Small-size and Large-size simulations and the seven sensitivity simulations (for the range of AeroCom models). The range (inter-model variability) of BC DRE by the AeroCom models was also defined as the difference between maximum and minimum values in the AeroCom model simulations[1]. Bond et al.[1] (Table 15) reported all-source BC direct radiative forcing (DRF) (difference between present-day and pre-industrial) of 0.20–0.36 W m$^{-2}$ for the AeroCom models. We assumed pre-industrial BC DRE was 20% of present-day BC DRE for these models to calculate BC DRE in the present-day based on the preindustrial (year 1850) to present-day (year 2000) fraction of BC emissions reported in Lamarque et al.[65] The maximum-minimum range is therefore 0.20 W m$^{-2}$ ((0.36–0.20)/0.8) for these models. The range of BC DRE from the AeroCom models derives from the different treatment of transport, transformation, deposition, size representation, parameters (e.g., density, refractive index), and interactions with radiation and cloud between the models.

Model intercomparison studies usually use 5–95% uncertainty ranges. Estimation of a 5–95% uncertainty range is useful when the number of models and/or simulations allows it. However, since this study is a single-model study where each simulation requires large amounts of computational cost, it is not practical to estimate the 5–95% uncertainty range. Rather, the range of maximum-minimum difference used in this study reflects that used by the AeroCom model intercomparison studies[34].

**Offline optical calculations**. Offline optical calculations were made to understand the causes of the large ranges of BC $AAOD_{550}$ and DRE in the Multiple-MS representation. We calculated $AAOD_{550}$ (5-year mean) from monthly outputs of the online simulations by using the Mie theory codes of Bohren and Huffman[69]. Similar to the online simulations, the core-shell treatment was used for internally-mixed BC particles and the well-mixed treatment was used for pure BC and BC-free particles. Aerosol optical depth at 550 nm ($AOD_{550}$) and $AAOD_{550}$ are almost the same between online simulations and offline calculations, as shown in Supplementary Figs. 10 and 11a: the slope is 0.99–1.01 and correlation coefficient ($R^2$) is greater than 0.996 for both $AOD_{550}$ and $AAOD_{550}$ (Supplementary Fig. 11a). The agreement of BC $AAOD_{550}$ is also very good between online simulations and offline calculations (Supplementary Table 3 and Supplementary Fig. 11b). Online simulations and offline calculations are not completely the same likely because of different temporal resolution of data. Two BC $AAOD_{550}$ values were calculated: (1) BC $AAOD_{550}$ including coating species with absorption enhancement (with the core-shell treatment) and (2) BC $AAOD_{550}$ excluding coating species (BC core only) without absorption enhancement (with the well-mixed treatment). The absorption enhancement factor ($E_{abs,coat}$) was defined as the ratio of these two $AAOD_{550}$ values.

We also conducted offline optical calculations assuming other BC MS assumptions (the dynamic effective medium approximation (DEMA)[40,70,71] and the Bruggeman mixing rule[71]) to confirm that our conclusions are robust. In the DEMA, the effective refractive index of BC-containing particles can be obtained from the effective dielectric constant ($\epsilon$), which is calculated from the following:[40,70,71]

$$(1 - f_{V\_BC}) \frac{\epsilon_{oth} - \epsilon}{\epsilon_{oth} + 2\epsilon} + f_{V\_BC} \frac{\epsilon_{BC} - \epsilon}{\epsilon_{BC} + 2\epsilon} + \frac{2\pi}{45} \left(\frac{2}{\lambda}\right)^2 (\epsilon_{BC} - \epsilon) \left[1 + \frac{5\epsilon}{2\epsilon_{BC} + 3\epsilon} + \frac{18\epsilon(\epsilon_{BC} - 2\epsilon)}{(\epsilon_{BC} + 2\epsilon)^2}\right] \int r^5 n(r) dr = 0 \quad (1)$$

where $f_{V\_BC}$ is the volume fraction of BC inclusions, $\epsilon_{BC}$ and $\epsilon_{oth}$ are the dielectric constant of BC and other species (including water), respectively, $\lambda$ is wavelength, and $r$ and $n(r)$ are the radius and size distribution of a BC inclusion, respectively. In the DEMA, BC inclusions were assumed to have monodisperse size distributions for each size and MS bin. We assumed each particle had multiple BC inclusions ($r = 50$ nm) if BC core size in a bin is greater than 50 nm in radius. In the Bruggeman mixing rule, the last (third) term in Eq. (1) is excluded by assuming infinitely small BC inclusions. In the Multiple-MS representation, the DEMA and the Bruggeman mixing rule have slightly lower sensitivity of $AAOD_{550}$ to emission size distributions than the core-shell treatment (Supplementary Fig. 8). However, all MS assumptions have high sensitivity in the Multiple-MS representation. Single-MS has almost negligible sensitivity to the size uncertainty for all MS assumptions. These results show that the main conclusions obtained in this study do not change by the choice of MS assumptions (used for internally-mixed BC particles) in.

**Estimation of DRE changes in the future**. DRE efficiency was defined for BC and total aerosol (sum of BC, organic aerosol, sulfate, and nitrate) as a change in DRE to per unit BC emission change. BC and total DRE efficiency was estimated for AN (fossil fuel + biofuel) and BB sources from the difference between the two simulations: one with the base emissions (for present-day) and the other with enhanced or reduced emissions (for future). Considering that global BC emissions from AN sources will decrease in the future[48], BC and total aerosol DRE efficiency for AN sources were calculated from the simulations with 50% reduced AN emissions for all emission species (gas and aerosol species). BC and total aerosol DRE efficiency for BB sources were calculated from the simulations with 50% enhanced BB emissions for all emission species because global BB BC emissions will increase in the future[49,50].

Total aerosol DRE and its efficiency in Fig. 6b, d were calculated from DREs scaled to the values of radiative forcing in Myhre et al.[3] In Myhre et al.[3], DRFs (relative to the year 1750) of BC and non-BC (sum of organic aerosol, sulfate, and nitrate) were 0.60 and −0.83 W m$^{-2}$, respectively. Since the magnitude of these values is 2.8 and 1.8 times greater than the DRF (relative to the year 1850) of BC (0.22 W m$^{-2}$) and non-BC (−0.46 W m$^{-2}$) in the Base-size simulation with the Multiple-MS representation, we used these values as scaling factors of BC and non-BC DREs for all simulation results in Fig. 6 b, d.

**Estimation of rapid adjustment of BC**. The effective radiative forcing of BC was estimated from the differences of the radiative flux at the top of atmosphere between the simulations with ten times enhanced BC emissions (×10 BC) and the simulations without BC emissions, following the method used in Stjern et al.[62] The rapid adjustment associated with BC was defined as the difference between the effective radiative forcing and DRE. We conducted 16-year online simulations with the Single-MS representation (for the Small-size, Base-size, and Large-size), and the latter 15 years were used for analysis. Multiple-MS is not available for this estimation because the 10 times enhancement of BC emissions only (without changing other species) leads to unrealistic calculations of BC aging processes and MS (therefore unrealistic estimates of BC DRE and rapid adjustment). The instantaneous cloud radiative effect of BC (×10 BC - without BC) is calculated based on the definition of Ghan[67] and shown in Supplementary Fig. 9 and Supplementary Table 4 for comparison.

**Code availability**. The codes used to conduct the analysis presented in this paper can be obtained by contacting the corresponding author (H.M.).

**Data availability**. Data used in Figs. 1, 3, 5 and 6 are available at http://has.env. nagoya-u.ac.jp/~matsui/data. Other data are available upon request from the corresponding author (H.M.)

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

## Acknowledgements

H.M. was supported by the Japan Society for the Promotion of Science (JSPS) Overseas Research Fellowships. This work was supported by the Ministry of Education, Culture, Sports, Science, and Technology and the Japan Society for the Promotion of Science (MEXT/JSPS) KAKENHI Grant Numbers JP26740014, JP17H04709, JP26241003, and JP16H01770 and MEXT Green Network of Excellence (GRENE) and Arctic Challenge for Sustainability (ArCS) projects. This work was also supported by the Environment Research and Technology Development Fund (2–1403, 2–1703) of Environmental Restoration and Conservation Agency. D.S.H. and N.M.M. acknowledge support from Atkinson Center for Sustainable Future at Cornell University. The authors thank Yutaka Kondo (National Institute of Polar Research in Japan), Nobuhiro Moteki (University of Tokyo), and National Oceanic and Atmospheric Administration (NOAA) Black Carbon Group for providing us their BC data of aircraft measurements.

## Author contributions

H.M. conceived and designed the research, performed model simulations and data analysis, and wrote the manuscript with collaboration with D.S.H. and N.M.M. H.M. and D.S.H. designed the uncertainty ranges of emissions size distributions and sensitivity simulations for BC mitigation impact. All authors suggested analysis, interpreted the data, discussed their implications, and contributed to the manuscript.

## Additional information

**Competing interests:** The authors declare no competing interests.

