## [Peer Review File · Nature Communications]

Reviewers' comments:

Reviewer #1 (Remarks to the Author):

Review of "Impact of particle size and mixing state diversity on estimates of black carbon mitigation" by Hitoshi Matsui, Douglas Hamilton and Natalie Mahowald.

General Comments and summary recommendation

This manuscript presents the results from a novel analysis which identifies that variations in the size distribution of emitted black carbon (BC) introduces a major uncertainty in the magnitude of the direct radiative effects from black carbon (BC). Furthermore, the paper explains how conventional aerosol models, which do not represent variations in BC mixing state, are under-representing this strong influence on the efficacy of BC radiative effects.

The authors reach these findings by applying the model described and evaluated in Matsui (2017) and Matsui and Mahowald (2017), to realistically resolve the different particle morphologies that exist and evolve following emission from different sources and subsequent chemical and microphysical transformations which occur during transport.

The analysis in the paper decomposes this "BC emissions-size-sensitivity effect" into its three main driving factors: BC lifetime, mass-absorption efficiency and core-shell absorption enhancement. In so-doing, the analysis then is able to illustrate why resolving the BC mixing state causes such an elevated sensitivity to variations in emissions particle size.

The analysis identifies that, in all three of these factors, resolving the internal mixing variations for the BC causes a stronger sensitivity to variations in emitted particle size. The 2nd and 3rd of the three steps both have particularly strong sensitivity to emissions-size in the internally mixed case, the BC forcing efficacy showing 40% difference between high and low emissions sizes, a factor 8 higher than in the single mixing state case (5%). The core-shell enhancement has a factor 3-4 larger variation in DRE when the mixing state variation is resolved. The authors are to be commended for providing such a comprehensive analysis to

demonstrate it is the combination of these latter two effects which explains why resolving the mixing state variations leads to such a heightened sensitivity to the emissions size.

The authors are also to be commended for carrying out a multi-model analysis of the AeroCom models to put their findings in context among the variation in BC DRE found by that range of models, and this is a particularly strong element of the paper.

The warming effects from anthropogenic black carbon emissions means the study's contribution in increasing our understanding of the uncertainty in model predictions of BC radiative effects are important to be communicated to the broader climate science community. Resolving variations in BC mixing state is cutting-edge modelling capability and this study is the first to assess how such variability affects BC climate forcing. For these reasons, the study's findings are certainly appropriate for publication in Nature Communications.

However, some parts of the text require substantial improvement, in particular the parts of the results and the important section that decomposes the drivers to identify why neglecting variations in BC-mixing-state causes the BC forcing uncertainty from the emissions-size-effect to be under-represented.

Also, whereas I can see that the Figures have the potential to succinctly summarize the main results (and importantly their driving reasons), the schematic diagram in Figure 2 needs to be changed to more effectively communicate the driving mechanisms identified in the analysis (see specific comments below).

I am therefore concluding that major revisions are required before publication in Nature Communications can occur, and have made a list of the specific revisions required. Despite major revisions being required, I think it likely the authors will be able to address these, and thereby revise the manuscript to ensure it communicates the important findings more effectively.

As well as the specific revisions requested, there are also one or two questions which the authors will need to provide explanation of, to clarify the basis on which the uncertainty ranges are being presented. I also have a suggestion for the authors to potentially make the study's findings more comparable to the ranges presented in Bond et al. (see revision 31).

Can the authors assign a quantitative probability estimate for the upper and lower size values they provide? If it could be determined what the probability

is that the emissions could be below the lower-range, and above the upper-range, that could then be straightforward to provide an equivalent 90% confidence interval for the different aspects, then matching the definition of the ranges in Bond et al. BC assessment.

Another aspect involves how the skewness of the BC DRE distribution should be interpreted, and whether the central estimate of the emissions size should be considered a median outcome in that particular metric, or whether indeed there is in fact any need to provide a central value -- perhaps a 90% confidence interval is sufficient for comparability to Bond14?

One final concern involves the choice of maximum and minimum BC sizes. Whereas resolving the mixing state variations would be expected to introduce a stronger variation between high and low emissions size for the core-shell enhancement effect, it was more surprising to me that the emissions size should change the radiative effect efficacy. And I raise here a concern about the comparability of the size range between the single MS and multiple MS cases.

I am concerned that the authors may inadvertently be conflating two issues here when they argue that simpler models, characterising only a single mixing state, must necessarily also require the BC to be emitted as 100% pure BC.

Although it is true to say that, with the exception of Bauer et al. (2008), global aerosol models (even the size-resolving models with aerosol microphysics) under-represent the diversity of BC mixing state, almost all the models resolve at least 2 mixing states in their representation of black carbon. Also, and more importantly, almost all the models co-emit particulate organic matter with the BC, treating even freshly emitted BC (either implicitly or explicitly) as an internal mixture of BC and POM. As a consequence, the emissions size distribution applied in the majority of these models will be very similar to the emissions size range labelled as being appropriate only for the multiple-mixing-state models.

I therefore find the 2nd of the three effects listed above, which finds factor of 8 higher sensitivity between low and high emissions size when mixing state variations are resolved, to be potentially misleading, as this effect is to do with how the BC emission is treated (externally mixed or internally mixed). Even models with only one BC mixing state resolved, will tend to co-emit BC also with POM or provide a size distribution for a "fresh soot" mode that implicitly reflects the properties of a core-shell particle.

I therefore request the authors must re-phrase their findings to explain that, for most single-MS models, it is only the 1st and 3rd of these factors that are relevant, the strong sensitivity from the 2nd effect (absorption efficiency/efficacy) only being applicable to the very small number of models that emit BC as an externally mixed pure BC particles (as well as only resolving one mixing state).

In addition to explaining that, the authors should either revise their uncertainty range, or provide an additional alternative range, which then is applicable for the single-mixing state models that co-emit the BC as internally mixed with POM (implicitly or explicitly) which I consider actually to be the majority of those single-MS models.

One other general point is that, in the decomposition into the three contributing mechanisms for how the emissions size affects the BC direct radiative effect efficacy, the variable names chosen to represent the 2nd and 3rd terms need to be revised, and the equals signs in equations 1, 2 and 3 replaced with proportional-to signs. Other improvements are also needed here such as to improve the variable naming and explanation of this novel and effective technique to decompose the BC aAOD into the 3 driving influences. For example "MAC_CORE" is applied to represent the absorption efficacy of the BC core, and the term "Eabs" applied to represent the effect of the coating around the BC core elevating the BC absorption (compared to pure BC core) when existing variables names with qualifying subscripts would much improve the accessibility of the approach being applied here. It is straightforward to rectify these issues however, and I have recommended in points 27 and 28 how to do that.

There are also a number of places where the text needs to be toned down, so as to be clear the BC mixing state is just one of several factors that affect BC DRE efficacy. For example the 2nd sentence "Partice size and mixing state determine the lifetime and solar absorption efficiency of BC" needs to be revised because other factors such as the nature of the precipitating clouds (e.g. ice water content) and surface roughness/vegetation also affect BC deposition processes. I understand of course that the authors are making the point that, those other things being equal, these effects play a primary role. But then the wording should be changed to express that more accurately, i.e. replace "determine" with "are both strongly influetial for" or "are both primary determining factors for". That way it is clear the two effects do not determine the effect on their own (it is not "determined" by them).

I suggest when revising the text, the lead author go through (with an English native speaker) the text and make sure that other similar statements are put into context

appropriately. I have listed a number of these in my specific revisions, but the co-authors should go through and together make any additional required amendments.

In summary, the paper requires major revisions, and a remedy or reply re: my point about a key element of the methodology, before it will be possible for the paper to proceed to publication in Nature Communications.

I am happy to review the paper again once these changes have been addressed, and I look forward to seeing the revised manuscript, and being able then to potentially approve the paper to then proceed to publication.

Specific Revisions

1) Lines 20-21: The authors need to add "Post-industrial increases in" or similar to be clear the warming effect is because black carbon concentrations have increased. There would be no climate effect if they had been present at the same concentration (aside from changes in their mixing state)..

2) Lines 21-22: The word "determine" is appropriate here re: the size and mixing states relationship to solar absorption (it is controlled by it). However, the lifetime of the BC does not just depend on its size and mixing state. For example the nature of the precipitation and of the clouds from which the precipitation is generated will also affect how effectively the BC is removed. Suggest to remove the words "lifetime and" and instead put that at the end as "and are also strongly influence how effectively they are removed".

3) Lines 22-23: this sentence is the authors opinion and is not appropriate as currently worded. I think this is the corollary the reader should take away from the paper -- but then this statement should be reserved for later in the Abstract, perhaps after the next three sentences that state the main results (at line 30). So that sentence could be reworded to follow on as:

"... total aerosol DRE changes from positive (warming effect) to negative (cooling effect). We therefore contend that aerosol models should either be improved to represent such BC mixing state variations, or else have their BC DRE uncertainty range amplified to account for the effects they are omitting."

4) Lines 23-24: Suggest to add, ", by running a variable-mixing-state aerosol microphysics model," or similar between "We show" and "that present-day" (it's

important, in my opinion, to mention the way that this is shown here). With that change you can remove "in a global aerosol model".

5) Line 27-28: Re-word "present-day BC DRE changes between 0.18 and 0.42 W/m²" to be clearer what you mean. Maybe just case of adding "by" between "changes" and "between"?

6) Line 42: As in my general comment above, the words "are controlling the burden" need to be revised because there are other factors too influencing these effects, so it is not just these two that are controlling. Suggest to revise "are controlling the burden (or lifetime) and heating effect..." to instead say "affect the BC removal rates and how strongly the surrounding air is heated." or similar.

7) Line 44: Replace "have various particle size and composition (mixing state, MS)..." with "have a range of particle sizes and mixing states (MS)..." -- I think that's clearer.

8) Line 45: Suggest to join this sentence up with the previous one, so have the previous sentence extended and revised from : "...even in the same air mass. These particles have different..." to instead say ", with the same air mass thereby having different... "

9) Line 48: Suggest to replace "However, these particle size and MS-dependent..." with "Such particle size and MS-dependent...." (better English), and then, again suggest to join up the current short final sentence of the para with this one, as "... sufficiently in most global aerosol models, which tend to have only 1 or 2 ..."

10) Line 53: Suggest to re-word as "changes in aerosol size distributions in emissions only" with "changes in assumed size distribution at emission, but only" (better English).

11) Lines 54-58: Need to make this sentence clear that this is what your study is showing. Suggest to re-word the start of the sentence changing "Changes in emissions..." to "Our simulations show that differences in emissions..." Then later in the sentence re-word "produce an unrecognized large uncertainty in BC DRE due to altering three BC properties simulated by the MS-resolved model: ..." with "produce a much larger uncertainty in BC DRE when the MS variations are resolved. The hitherto unrecognised effect occurs because the MS-resolving model simulates three key properties of the BC differently when the MS variations are simulated: ..."

12) Line 61: Change the start of this 1st sentence of the paragraph to say "We carry out model experiments to quantify the importance of resolving the

size-resolved BC mixing state by comparing...." (delete the specifics of the "multiple MS categories (here 8 MS categories) is" as this is in the methods.

13) Lines 64-67: Suggest to start this sentence from "Since all aerosol processes" with "In these model experiments, all aerosol processes..." and then replace "treatment, Multiple-MS can calculate.." with "treatment, with Multiple-MS a benchmark simulate that realistically calculates.." -- with that change of wording can delete "in detail" from the end of the sentence (it's better to state that this is done realistically than say it's done in detail).

14) Lines 67-68: Re-word "In contrast, Single-MS cannot calculate these properties because all particles are assumed to be thickly-coated BC particles." with "The Single-MS run is designed to represent the simplified BC treatment used in most models, all particles assumed to have the same coating thickness for BC particles."

15) Lines 68-69: Re-word "We primarily use simulations with the Small, Base and Large emission size...." as "With each model set-up, experiments are carried out with Small, Base and Large emissions size...."

16) Lines 70-73: Explain to the reader the reason why you carry out this AeroCom analysis, by changing the start of this sentence from "We also use seven additional simulations, which are based on a range of emissions size distributions used in the Aerosol Comparisons..." with "To put our results in the context of current assumptions in models, we also carry out seven additional simulations applying the emissions sizes used by models participating in the Aerosol Comparisons..." with this change of wording you can then delete the "models, to support the conclusions from the Small-, Base- and Large-size simulations." (just finish the sentence after citing the reference 21).

17) Lines 75-77: Need to change the wording of this sentence -- suggest to re-word as "Our results demonstrate the importance of accurately identifying the emissions size distribution, but additional simulations (see Supplementary Discussion S1) show the results are not sensitive to the exact mixing state assumed at emission."

18) Lines 79-83: Move the citing of references 22 and 23 to be earlier in the sentence, to come just after "Our previous studies" (and delete the "also").

19) Lines 83-87: This is too much detail at this stage -- this text and that in lines 75-83 -- needs to come after the main results have been presented, in a Discussion section of the manuscript. Please go through the manuscript and think through the sequencing of the text -- the Introduction certainly needs a couple of additional sentences at the start to motivate the importance of the study with reference to other previous related studies. And moving this section to later could accommodate that addition whilst keeping the Introduction the same length.

20) Line 91 -- Add an introductory sentence to first identify/establish the rationale behind the analysis you are carrying out -- to assess the importance for the radiative effects of BC of resolving its mixing state through each of the three aspects identified earlier.

21) Line 92,93 -- the fact that the single-MS representation has only 19% variation compared to 32% resolved in the multiple-MS case is important and could be re-iterated/re-interpreted in the Abstract as the single-MS missing 68% (32/19-1) of the variability from the different size distribution (you could cite an example of this arising from different types of combustion sources for example). This is only for the BC lifetime, and it could be combined with the others to provide a summary in the Abstract to give a quantitative measure to report objectively to allow the reader to determine how important it is for models to resolve the effect.

22) Lines 93-96 -- The authors need to qualify their assessment however with some reference to the fact that many of the single-MS models will likely assign the particles/mode/size-bin properties that implicitly provide a hygroscopicity and CCN-activity that will likely represent a low-coated BC particle rather than a pure BC particle (even if that is what the model actually resolves).

23) Line 95 -- it is likely not to be obvious to some readers that a larger variability in CCN-activity necessarily results in a larger variation in BC lifetime. Perhaps it's just a case of removing "This is because Multiple-MS resolves..." to instead say "This stronger BC lifetime sensitivity results from Multiple-MS resolving..." (and also add a comma after "thinly coated BC"), and additionally adding a sentence after that to explain the link to BC lifetime,

suggest something like:

"In the CAM5-chem/ATRAS model, higher CCN activity translates into a shorter BC lifetime since wet removal is its dominant sink (e.g. Textor et al., 2006; Matsui, 2017)."

24) Lines 93-96 -- Has this sensitivity to BC emissions size distribution been assessed before? Can these numbers for the emissions-size sensitivity for BC lifetime be compared to those derived from other studies? For example Reddington et al. (2013), using a sectional global aerosol microphysics model with 3 distributions of bins that contain BC, present results which illustrate similar sensitivity of simulated surface BC to assumptions on emissions size distribution, finding higher surface BC for larger assumed size than smaller size (e.g. as 76 and 122 ng/m³ on the EUCAARI campaign mean). The model applied in that analysis did not not resolve the effect of the mixing state on "CCN activity", but did resolve variations in the rate of "condensation-ageing" across the particle size range, and identified associated slower ageing as the reasons for the higher BC in the small emissions size case. The authors need to add some reference to this work either here or in some discussion prior to the conclusions of the paper. The analysis in Reddington et al. (2013) also compared the BC coating thickness (across the size range) to that measured by the SP2 instrument, and the authors should refer to how this compares also in their model, or if they have evaluated it in their papers, explain how (in broad terms) their simulated size resolved BC coating thickness compares to that shown in that paper for the simulations there with that particular 3-mixing-state sectional approach. Suggest to add 1 sentence to this paragraph referring to these results.

25) Lines 97-105 -- The acronym AAOD is used in the paper to denote absorption aerosol optical depth, but I recommend to change this "AAOD" to "aAOD" since the AOD is an established three-letter-acronym (TLA), the absorption then being a qualifying term that should be in lower case ahead of that established TLA. I would also recommend the authors always provide the wavelength in subscript whenever the acronym aAOD or AOD is used in papers. So change "AAOD" to "aAOD" always with subscript 550 if this is the only wavelength assessed.

26) Line 116 -- replace "Figs 1e-1f" with "Figs 1e-f".

27) Lines 121-127 -- the decomposition of the BC aAOD₅₅₀ into the three

driving influences is a novel and effective way to explain why the single-MS approach is under-representing the uncertainty in BC DRE. However, the equations 1, 2 and 3 on lines 122-124 are not really correct, as each one is currently stating that the numerical value of the term on the left-hand-side can be recovered simply by multiplying the terms on the right-hand side. I think the authors are really saying that the terms on the left-hand side are proportional to the product/ratio of terms shown on the right-hand side, in which case the problem is easily remedied by replacing the equals sign instead with a "proportional-to" sign. Also, please remove "(BC + nonBC)" from the left-hand side here as the term "BC AAOD" (or BC aAOD550 as needs to be written) is already being clear it is a "speciated AOD" (just the component of the absorption-AOD that occurs because of the BC component).

28) The variable names for the terms MACcore and Eabs in equations 1, 2 and 3 also need to be improved as the paper is revised.

For the absorption efficacy term, the authors should follow the variable naming from Seinfeld and Pandis, "Atmospheric Chemistry and Physics: From Air Pollution to Climate Change.....",

which has the variable "Q" subscript "abs" used for the mass absorption cross-section.

The authors could specify this as "Q" subscript "abs,core" to be clear it is applied to the BC core in this case. For the Eabs term, really, this subscript "abs" is not quite the correct qualifying term (on its own at least), as the effect is really, as explained on page 6, that the coating effect promotes absorption over that which would be the case for a pure BC particle with the same amount of black carbon mass.

I therefore recommend to change the name of the term

"E" subscript "abs" to instead be with "E" subscript "coat" or else subscript "abs,coat" or "abs,core-shell". That way it will be immediately clear to the reader that that term is representing the enhancement of the absorption due to the coating or core-shell effect.

In addition to the above naming changes, I suggest an overline or hat symbol ("^") is added to each term because these terms are not providing any specific gridbox value, but are rather an average or central value, representative of how the effect manifests in the column-integrated BC-attributed absorption (the BC absorption optical depth).

29) Lines 127 to 130 -- the sentence beginning on line 127 needs to be moved to the start of this section (i.e. begin the paragraph with this), the text after "The three terms in the right side of the equation" deleted and the remainder of the sentence then following after "to three effects" in

the current 1st sentence, also with an initial brief few words to explain why this is being done such as "To explain why single-MS approaches are under-representing BC DRE uncertainty, we decompose BC aAOD550 into three component elements:" then add the text from the 2nd half of the current lines 127-130.

30) Line 131 -- "...since the mass flux of emissions is held constant." Suggest to reword "is held constant" to "is the same in all simulations". Also, need to explain that the change in size results in much more particles being emitted, suggest to add "the number of particles emitted increasing as emission size decreases" or similar to briefly remind the reader of this.

30) Line 133-134 -- The authors have a 1-sentence paragraph here which looks like somehow how the submitted manuscript is missing some key text. Suggest to add 2 or 3 sentences to complete this paragraph providing explanation of how the emissions-size influences the BC burden/lifetime in the CAMS5-chem/ATRAS model and how other models do this differently. This is one of the key points that provide the explanation for why the single-MS approach under-represents the BC burden/lifetime sensitivity to emissions-size.

31) Figure 1 -- the authors have explained in the text (lines 360-362) that the uncertainty ranges presented here are not exactly the same as in the 5 to 95% range given in the Bond et al. (2013) BC assessment. Although that is true, if a probability could be given for the limit values in particle size considered (e.g. the upper size limit is 90% or 95%) then the ranges given could be converted into a comparable confidence interval. Could the authors give some statement about their expert judgement for what the small-size and large-size represent? Can this be considered probabilistically? Perhaps it would not be appropriate to convert into the range because the authors are really testing the sensitivity (so it's a sensitivity not an uncertainty). If that is the case then the authors should add a sentence explaining why the range given cannot be similarly converted into an uncertainty range to be compared on a similar basis.

32) Figure 2 -- I really like this diagram as the illustration really helps the reader understand visually what is being explained in the text. However, I don't think the idea to have the blue and red arrows

showing positive and negative responses really works (at least not as currently shown). Why are the arrows to the right of the boxes in red and the arrows to the left in blue? And why are there "(+/-)" signs all in red above each box. Also, the left-hand box says "Particle size (emission, atmosphere)" but that text would be better understood by the reader if it was changed to "Emissions particle size". Certainly it is not necessary to state "atmosphere".

Also, it is my opinion that this Figure is fine to just illustrate the 3 driving influences/components of the BC aAOD550 that the authors decomposed in equations 1, 2 and 3. With this in mind, I suggest that, when revising the paper, the authors simplify this Figure, having just one arrow in each of the three drivers/components that are connecting up from the emissions particle size on the left to the aAOD550 on the right.

33) Figure 3 -- replace "MACcore" with Q-subscript-abs,core as in comment 29. Also, on line 674 the text "theoretical calculations" needs to be better qualified with the mention of the method used to translate the different mixing state into absorption cross-section.

References

Matsui, H.:

Development of a global aerosol model using a two-dimensional sectional method:

1. Model design,

J. Adv. Model. Earth Syst., 9, 1921-1947, doi:10.1002/2017MS000936.

Matsui, H. and Mahowald, N.:

Development of a global aerosol model using a two-dimensional sectional method:

2. Evaluation and sensitivity simulations,

J. Adv. Model. Earth Syst., 9, 1887-1920, doi:10.1002/2017MS000937.

Reddington, C. L., G. McMeeking, G. W. Mann, H. Coe, M. G. Frontoso,

D. Liu, M. Flynn, D. V. Spracklen and K. S. Carslaw:

The mass and number size distributions of black carbon aerosol over Europe

Atmos. Chem. Phys., 13, 4917-4939, 2013.

Textor, C., M. Schulz, S. Guibert, S. Kinne, Y. Balkanski, S. Bauer, T. Berntsen,

T. Berglen, O. Boucher, M. Chin, F. Dentener, T. Diehl, R. Easter, H. Feichter, D. Fillmore, S. Ghan, P. Ginoux, S. Gong, A. Grini, J. Hendricks, L. Horowitz, P. Huang, I. Isaksen, T. Iversen, S. Kloster, D. Koch, A. Kirkevag, J. E. Kristjansson, M. Krol, A. Lauer, J. F. Lamarque, X. Liu, V. Montanaro, G. Myhre, J. Penner, G. Pitari, S. Reddy, Å. Seland, P. Stier, T. Takemura and X. Tie
Analysis and quantification of the diversities of aerosol life cycles within AeroCom,
Atmos. Chem. Phys., 6, 1777–1813, 2006.

Reviewer #2 (Remarks to the Author):

Review of Matsui et al, Impact of particle size and mixing state diversity on estimates of black carbon mitigation.

The paper investigates black carbon mixing state effects on climate.

L 37: BC has the largest human made impact on climate change. Most of the BC in the atmosphere is from biomass burning, thus a large portion is natural. Overall I would say sulfate has the largest man made impact on climate when looking at aerosol species.

L 49 The paper claims that BC mixing state effects are not represented in global climate models. This claim is not true as all models that include aerosol microphysical schemes represent this process in some degree. Some models do even simulate those processes in great detail, such as Jacobson 2001 and Bauer et al 2008.

L83: It has been clearly demonstrated that core-shell representation overestimate BC absorption enhancement. The results presented in the supplementary material disagree with Adachi et al (2010) and Cappa et al 2012

SI: The paper compares the simulation to global mean budget numbers of other aerocom models, but is completely lacking any comparisons to observations. Global mean numbers are not a good measure to investigate locally and regional acting aerosol effects.

L117: Most or all climate models use BC enhancement factors in order to parameterize BC mixing state effects. Even so the process is not physically resolved in all models, it is not absent in the overall results.

The impact of BC emission size on BC aerosol forcing have been investigated in Bauer et al (2010)

L222 The impact of biofuel vs fossil fuel emissions have been discussed in Jacobson (multiple papers see below) and Chen et al (2010)

A model description Page 12 to 20 should not be published in a Nature paper.

My overall comment is that this is a very nice and interesting study. But not suitable for the journal submitted here, due to the lack of new and original material. The paper lacks key publications on the topic of BC mixing state and thus its findings are not completely unique. I would recommend to include a data evaluation part in this paper and resubmit it to a different journal, such as ACP or JGR.

References:

Adachi, K., Chung, S. H. & Buseck, P. R. Shapes of soot aerosol particles and implications for their effects on climate. *J. Geophys. Res.* 115, D15206 (2010).

Bauer, S.E., S. Menon, D. Koch, T.C. Bond, and K. Tsigaridis, 2010: A global modeling study on carbonaceous aerosol microphysical characteristics and radiative forcing. *Atmos. Chem. Phys.*, 10, 7439-7456, doi:10.5194/acp-10-7439-2010.

Cappa et al 2012, Radiative Absorption Enhancements Due to the Mixing State of Atmospheric Black Carbon, *Science* 31 Aug 2012

Chen, W. T., Lee, Y. H., Adams, P. J., Nenes, A., and Seinfeld, J. H. (2010) "Will black carbon mitigation dampen aerosol indirect forcing?," *Geophys. Res. Lett.*, 37(L09801): doi:10.1029/2010GL042886.

Jacobson, M. Z., GATOR-GCMM: A global through urban scale air pollution and weather forecast model. 1. Model design and treatment of subgrid soil, vegetation, roads, rooftops, water, sea ice, and snow. *J. Geophys. Res.*, 106, 5385-5402, 2001.

Jacobson, M. Z., Effects of biomass burning on climate, accounting for heat and moisture fluxes, black and brown carbon, and cloud absorption effects (*J. Geophys. Res.*, 2014)

Jacobson, M. Z., Cloud absorption effects and boomerang curves: Optical properties of black carbon, tar balls, soil dust in clouds and aerosols, *J. Geophys. Res.*, 2012

Jacobson, M. Z., Short-term effects of controlling fossil-fuel soot, biofuel soot and gases, and methane on climate, the Arctic, and health, *J. Geophys. Res.*, 2010

Jacobson, M. Z., Effects of Externally-Through-Internally-Mixed Soot Inclusions Within Clouds

and Precipitation on Global Climate, J. Phys. Chem. A, 2006.

Reviewer #3 (Remarks to the Author):

Review of manuscript by Matsui et al., entitled 'Impact of particle size and mixing state diversity on estimates of black carbon mitigation'. The manuscript investigates the uncertainty associated with emission size of black carbon (BC) and mixing of BC with other aerosol types. The main finding is that a more sophisticated aerosol size and mixing scheme shows a much larger uncertainty range in the radiative forcing of the direct radiative effect of BC relative to standard schemes applied in global climate models. A large number of previous studies have investigated the mixing state of BC, but the investigation of emission size on the mixing state of BC is novel. Given all the policy discussions of BC mitigation the scientific topic addressed in the manuscript by Matsui et al. is highly important. It is therefore vital that improved knowledge and uncertainties on BC climate effect are published in the top scientific journals. In principle, I would recommend the manuscript by Matsui et al. for Nature Communication. However, the current version is missing some important aspect of the BC climate effect in terms of mitigation. Below are my two main concerns as well as some minor recommendation for improvements.

1) The manuscript lacks inclusion of the rapid adjustment associated with BC (the term semi-direct effect has previously often been used, but rapid adjustment is a broader term than semi-direct effect often associated with BC induced cloud changes. The rapid adjustment is shown to be highly important for BC in multi-model climate simulations (Stjern et al., 2017) and explains a low temperature response from BC (Baker et al., 2015; Samset et al., 2016). Since the global mean temperature change from BC in global climate models is lower than expected the rapid adjustment is essential to include to provide the full picture of BC for mitigation purposes. In several climate models the offset by the rapid adjustment of the direct radiative effect is almost 50% (Stjern et al., 2017), highlighting this as a necessary effect to include. Furthermore, the rapid adjustment of BC is shown in a single-model to vary inversely with the strength of the direct radiative effect, given a net climate impact which is less sensitive to uncertainties associated with BC (Hodnebrog et al., 2014). The effective radiative forcing (ERF) and thereby the rapid adjustment should be fairly easy to extract from the performed simulations in the manuscript by Matsui et al. Since the BC signal is rather small, expanding the number of years of simulations may be needed.

2) The climate effect of BC is uncertain because of many factors influencing the forcing and few constraints from observations. Anyway, we have two constraints from observations on the vertical profile of BC and mass absorption coefficient (MAC). Reference number 10 in Matsui et al. shows vertical profiles of BC compared to observations. Since the lifetime of BC changes in

model simulations, I recommend to show how the vertical profile of BC compares with observations. A standard finding in the global models is that the abundance of BC is overestimated in the upper troposphere compared to observations (Samset et al., 2014; Wang et al., 2014a) and it would be valuable to see if any of the model simulations performed in this study compares better or worse with the observations and possibly be used as a constrain. There are uncertainties in the measurements and large spatial and temporal variations of MAC, but a wide range of measurements points to a majority of MAC somewhere between 7 and 13 m²/g at 550 nm (Bond and Bergstrom, 2006; Bond et al., 2013; Zanatta et al., 2016). Are the simulated MAC values within the range from observations?

Minor comments:

Line 34-36: Bond et al. (ref 1) highlight BC as the second largest positive climate forcing in terms of emissions. Note that there is a difference between describing this in terms of emissions and in abundance as the latter is implied in the manuscript. For methane the emission based forcing is much larger than the abundance based forcing. There is several published papers since 2013 indicating that Bond et al. overestimate the BC forcing. Wang et al. (2014b) shows that high resolution modelling reduce substantially the difference between model simulation of AOD and AERONET data, indicating that AERONET stations generally is located close to BC emission sources and overemphasizing the difference between AERONET and global models. Samset et al. (2014); Wang et al. (2014a) shows that global models generally has too long BC lifetime and overestimate the BC forcing. In addition the rapid adjustment of BC seems much more important than provided in Bond et al (2013), see point 1 above. Given the more recent findings on BC, I don't think highlighting BC as the second most important forcer is correct and think you should refrain from repeating this. Highlighting the uncertainty on the other hand is fine.

Introduction: The importance of mixing for cloud condensation nuclei (CCN) activity is mentioned several times. On line 260 it is described that aerosol-cloud interaction is included together with aerosol-radiation interaction (direct radiative effect). It would be nice to add a table separating the aerosol-cloud interaction from the aerosol-radiation interaction (as well as including the rapid adjustment as suggested in point 1 above).

Line 109-110: I have problem finding the uncertainty range of 0.20 Wm⁻² from the AeroCom models in reference 1. This value is much higher than from the original AeroCom paper.

Line 127-132: It could be mentioned that BC burden and Eabs is related, this is shown in Boucher et al. (2016).

Line 333-336 & line 360-362: The uncertainty range here taken as difference between maximum and minimum values is not a standard approach. This needs to be confirmed to be similar to

approaches that are more standard.

Line 338-342: This is very unclear. Do you describe anthropogenic BC emission prior to 1750 (or 1850)?

Line 348-360: Given the concern that Bond et al (ref 1) overestimate the forcing of BC (including the DRE), I recommend to not scale your results to the Bond et al estimate.

Line 416-418: The values quoted from (Myhre et al., 2013) (ref 3) is DRE (RF ari in IPCC AR5 terminology) and not ERF as given in the text.

Baker, L. H., Collins, W. J., Olivie, D. J. L., Cherian, R., Hodnebrog, Ø., Myhre, G. and Quaas, J.: Climate responses to anthropogenic emissions of short-lived climate pollutants, *Atmos. Chem. Phys.*, 15(14), 8201-8216, 2015.

Bond, T. C. and Bergstrom, R. W.: Light absorption by carbonaceous particles: An investigative review, *Aerosol Science And Technology*, 40(1), 27-67, 2006.

Bond, T. C., Doherty, S. J., Fahey, D. W., Forster, P. M., Berntsen, T., DeAngelo, B. J., Flanner, M. G., Ghan, S., Karcher, B., Koch, D., Kinne, S., Kondo, Y., Quinn, P. K., Sarofim, M. C., Schultz, M. G., Schulz, M., Venkataraman, C., Zhang, H., Zhang, S., Bellouin, N., Guttikunda, S. K., Hopke, P. K., Jacobson, M. Z., Kaiser, J. W., Klimont, Z., Lohmann, U., Schwarz, J. P., Shindell, D., Storelvmo, T., Warren, S. G. and Zender, C. S.: Bounding the role of black carbon in the climate system: A scientific assessment, *Journal of Geophysical Research-Atmospheres*, 118(11), 5380-5552, 2013.

Boucher, O., Balkanski, Y., Hodnebrog, Ø., Myhre, C. L., Myhre, G., Quaas, J., Samset, B. H., Schutgens, N., Stier, P. and Wang, R.: Jury is still out on the radiative forcing by black carbon, *Proceedings of the National Academy of Sciences*, 113(35), E5092-E5093, 2016.

Hodnebrog, Ø., Myhre, G. and Samset, B. H.: How shorter black carbon lifetime alters its climate effect, *Nat Commun*, 5, 5065, 2014.

Myhre, G., Shindell, D., Bréon, F.-M., Collins, W., Fuglestedt, J., Huang, J., Koch, D., Lamarque, J.-F., Lee, D., Mendoza, B., Nakajima, T., Robock, A., Stephens, G., Takemura, T. and Zhang, H., Anthropogenic and Natural Radiative Forcing. In: *Climate Change 2013: The Physical Science Basis. Contribution of Working Group I to the Fifth Assessment Report of the Intergovernmental Panel on Climate Change*. T. F. Stocker, D. Qin, G.-K. Plattner, M. Tignor, S. K. Allen et al. (Editors), Cambridge University Press, Cambridge, United Kingdom and New York, NY, USA, pp. 659-740, 2013.

Samset, B. H., Myhre, G., Forster, P. M., Hodnebrog, Ø., Andrews, T., Faluvegi, G., Fläschner, D., Kasoar, M., Kharin, V., Kirkevåg, A., Lamarque, J. F., Olivié, D., Richardson, T., Shindell, D., Shine, K. P., Takemura, T. and Voulgarakis, A.: Fast and slow precipitation responses to individual climate forcings: A PDRMIP multimodel study, *Geophysical Research Letters*, 43(6), 2782-2791, 2016.

Samset, B. H., Myhre, G., Herber, A., Kondo, Y., Li, S. M., Moteki, N., Koike, M., Oshima, N., Schwarz, J. P., Balkanski, Y., Bauer, S. E., Bellouin, N., Bernsten, T. K., Bian, H., Chin, M., Diehl, T., Easter, R. C., Ghan, S. J., Iversen, T., Kirkevåg, A., Lamarque, J. F., Lin, G., Liu, X., Penner, J. E.,

Schulz, M., Seland, Ø., Skeie, R. B., Stier, P., Takemura, T., Tsigaridis, K. and Zhang, K.: Modelled black carbon radiative forcing and atmospheric lifetime in AeroCom Phase II constrained by aircraft observations, *Atmos. Chem. Phys.*, 14(22), 12465-12477, 2014.

Stjern, C. W., Samset, B. H., Myhre, G., Forster, P. M., Hodnebrog, Ø., Andrews, T., Boucher, O., Faluvegi, G., Iversen, T., Kasoar, M., Kharin, V., Kirkevåg, A., Lamarque, J.-F., Olivié, D., Richardson, T., Shawki, D., Shindell, D., Smith, C. J., Takemura, T. and Voulgarakis, A.: Rapid Adjustments Cause Weak Surface Temperature Response to Increased Black Carbon Concentrations, *Journal of Geophysical Research: Atmospheres*, 122(21), 11,462-11,481, 2017.

Wang, Q. Q., Jacob, D. J., Spackman, J. R., Perring, A. E., Schwarz, J. P., Moteki, N., Marais, E. A., Ge, C., Wang, J. and Barrett, S. R. H.: Global budget and radiative forcing of black carbon aerosol: Constraints from pole-to-pole (HIPPO) observations across the Pacific, *Journal of Geophysical Research-Atmospheres*, 119(1), 195-206, 2014a.

Wang, R., Tao, S., Balkanski, Y., Ciais, P., Boucher, O., Liu, J. F., Piao, S. L., Shen, H. Z., Vuolo, M. R., Valari, M., Chen, H., Chen, Y. C., Cozic, A., Huang, Y., Li, B. G., Li, W., Shen, G. F., Wang, B. and Zhang, Y. Y.: Exposure to ambient black carbon derived from a unique inventory and high-resolution model, *Proceedings of the National Academy of Sciences of the United States of America*, 111(7), 2459-2463, 2014b.

Zanatta, M., Gysel, M., Bukowiecki, N., Müller, T., Weingartner, E., Areskoug, H., Fiebig, M., Yttri, K. E., Mihalopoulos, N., Kouvarakis, G., Beddows, D., Harrison, R. M., Cavalli, F., Putaud, J. P., Spindler, G., Wiedensohler, A., Alastuey, A., Pandolfi, M., Sellegri, K., Swietlicki, E., Jaffrezo, J. L., Baltensperger, U. and Laj, P.: A European aerosol phenomenology-5: Climatology of black carbon optical properties at 9 regional background sites across Europe, *Atmospheric Environment*, 145(Supplement C), 346-364, 2016.

Response to reviewer #1

NCOMMS-17-30621-T: “Impact of particle size and mixing state diversity on estimates of black carbon mitigation” by H. Matsui et al.

We thank the reviewer very much for reading the paper carefully and giving us valuable comments. We revised the paper by taking into account the reviewer’s comments. In this revision, we revised the main text based on all important comments by the reviewer. We added offline optical calculations to support our conclusions on the BC core absorption effect. We also revised Figure 2 considering the reviewer’s comment.

Detailed responses to individual comments and suggestions are given below.

Reviewer’s comment:

However, some parts of the text require substantial improvement, in particular the parts of the results and the important section that decomposes the drivers to identify why neglecting variations in BC-mixing-state causes the BC forcing uncertainty from the emissions-size-effect to be under-represented.

Response:

This comment is related to the specific comments 20)-24) and 29)-30). We revised the text following these reviewer’s comments. Please see the response to individual specific comments shown below. We believe the manuscript was much improved thanks to the reviewer’s suggestions.

Reviewer’s comment:

Also, whereas I can see that the Figures have the potential to succinctly summarize the main results (and importantly their driving reasons), the schematic diagram in Figure 2 needs to be changed to more effectively communicate the driving mechanisms identified in the analysis (see specific comments below).

Response:

We simplified Figure 2 (Figure 4 in the revised manuscript) by removing (+/-) signs and the description of “pure-BC, thinly-coated BC”, “BC-free”, and “Thickly-coated BC” and by changing “Particle size (emission, atmosphere)” to “Emissions particle size” (as the reviewer suggested). We added the explanation of blue and red arrows to the figure caption to clarify their meanings: “Specifically, decreasing particle sizes at emission (e.g. Small-size simulation) increases BC lifetime, BC core absorption, and absorption enhancement, and these enhancements increase absorption aerosol optical depth at 550 nm (AAOD₅₅₀) and DRE of BC in the Multiple-mixing-state representation.” (Lines 807-810). We want to retain the information of DRE and their values because understanding BC DRE responses is the main purpose of these analyses.

Reviewer's comment:

As well as the specific revisions requested, there are also one or two questions which the authors will need to provide explanation of, to clarify the basis on which the uncertainty ranges are being presented. I also have a suggestion for the authors to potentially make the study's findings more comparable to the ranges presented in Bond et al. (see revision 31).

Can the authors assign a quantitative probability estimate for the upper and lower size values they provide? If it could be determined what the probability is that the emissions could be below the lower-range, and above the upper-range, that could then be straightforward to provide an equivalent 90% confidence interval for the different aspects, then matching the definition of the ranges in Bond et al. BC assessment.

Another aspect involves how the skewness of the BC DRE distribution should be interpreted, and whether the central estimate of the emissions size should be considered a median outcome in that particular metric, or whether indeed there is in fact any need to provide a central value -- perhaps a 90% confidence interval is sufficient for comparability to Bond14?

Response:

We deleted comparisons with the range in Bond et al. BC assessment (Figure 1f), considering a comment by Reviewer #3. This deletion does not change our main conclusions because we mainly use Figure 1e (comparisons with the AeroCom models; Figure 3 in the revised manuscript) for comparisons with the ranges by other aerosol models.

As described in Lee et al. (2013), carbonaceous aerosol particle size is a very uncertain parameter because the size of combustion particles from fossil fuels depends on the details of the source (e.g. fuel and industry type and age of the facility) and the size from biomass burning and wildfires depends on the burning efficiency of the fuel and its properties (e.g., moisture content and fuel type). Their ranges of emission particle sizes correspond to the maximum-minimum ranges of median diameters of emissions assumed in the AeroCom models (Supplementary Table 2). The range of

median diameters for fossil fuel sources is 30-80 nm in both Lee et al. (2013) and the AeroCom models. The range for biofuel and biomass burning sources in Lee et al. (2013) (50-200 nm) are slightly larger than the AeroCom models (30-150 nm), but a recent study suggests carbonaceous biomass burning emission could have 300 nm in diameter in maximum (Regaryre et al., 2018).

We added these sentences to Methods (Lines 436-446) in the revised manuscript.

Reviewer's comment:

One final concern involves the choice of maximum and minimum BC sizes. Whereas resolving the mixing state variations would be expected to introduce a stronger variation between high and low emissions size for the core-shell enhancement effect, it was more surprising to me that the emissions size should change the radiative effect efficacy. And I raise here a concern about the comparability of the size range between the single MS and multiple MS cases.

I am concerned that the authors may inadvertently be conflating two issues here when they argue that simpler models, characterising only a single mixing state, must necessarily also require the BC to be emitted as 100% pure BC.

Although it is true to say that, with the exception of Bauer et al. (2008), global aerosol models (even the size-resolving models with aerosol microphysics) under-represent the diversity of BC mixing state, almost all the models resolve at least 2 mixing states in their representation of black carbon. Also, and more importantly, almost all the models co-emit particulate organic matter with the BC, treating even freshly emitted BC (either implicitly or explicitly) as an internal mixture of BC and POM. As a consequence, the emissions size distribution applied in the majority of these models will be very similar to the emissions size range labelled as being appropriate only for the multiple-mixing-state models.

I therefore find the 2nd of the three effects listed above, which finds factor of 8 higher sensitivity between low and high emissions size when mixing state variations are resolved, to be potentially misleading, as this effect is to do with how the BC emission is treated (externally mixed or internally mixed). Even models with only one BC mixing state resolved, will tend to co-emit BC also with POM or provide a size distribution for a "fresh soot" mode that implicitly reflects the properties of a core-shell particle.

I therefore request the authors must re-phrase their findings to explain that, for most single-MS models, it is only the 1st and 3rd of these factors that are relevant, the strong sensitivity from the 2nd effect (absorption efficiency/efficacy) only being applicable to the very small number of models that emit BC as an externally mixed pure BC particles (as well as only resolving one mixing state).

In addition to explaining that, the authors should either revise their uncertainty range, or provide an additional alternative range, which then is applicable for the single-mixing state models that co-emit the BC as internally mixed with POM (implicitly or explicitly) which I consider actually to be the majority of those single-MS models.

Response:

We thank the reviewer for providing us this important comment. The cause of

the higher sensitivity for the 2nd (BC core absorption) effect in our Multiple-MS model simulation is not the treatment of BC emissions, but rather the treatment of BC-free particles. To help clarify this point for the reader we have conducted additional offline optical calculations (using the Multiple-MS results) and added these new results to the main text as follows.

“Since many global aerosol microphysics models consider 2 MS categories (hydrophobic and hydrophilic), we conducted offline optical calculations, assuming 2 and 3 MS categories, using the results of Multiple-MS simulations to further highlight the importance of representing multiple BC MS. In the 2-MS treatment, we consider hydrophobic and hydrophilic BC groups (BC mass fractions are assumed as 0.9-1.0 and 0-0.9, respectively) from the 8 MS information in Multiple-MS. In the 3-MS treatment, hydrophobic BC, hydrophilic BC, and BC-free groups (BC mass fractions are assumed as 0.9-1.0, 0.0001-0.9, and 0-0.0001, respectively) are considered. The Small/Large ratios in the 2-MS treatment are 1.04 and 0.94 for the BC core absorption and absorption enhancement effects, respectively, suggesting that the 2-MS treatment cannot capture the sensitivity of these two effects as simulated by Multiple-MS (Small/Large ratios are 1.41 and 1.15, respectively, in Multiple-MS). The Small/Large ratios in the 3-MS treatment are 1.24 and 1.19 for the two effects, showing a much increased sensitivity for the two effects by the addition of resolving BC-free particles. However, the sensitivity for the BC core absorption effect is still underestimated (by 41%) compared with the Multiple-MS treatment. An important component of most global aerosol models with 2 BC MS categories is the chosen threshold value to distinguish between hydrophobic and hydrophilic BC. This aging threshold term is highly variable, and often modified in order to better correlate simulated BC properties with observations. Therefore, resolving multiple BC MS categories removes the dependence of choosing a threshold and represents the diversity of CCN activity and absorption enhancement of BC-containing particles more realistically.” (Lines 256-277).

Reviewer’s comment:

One other general point is that, in the decomposition into the three contributing mechanisms for how the emissions size affects the BC direct radiative effect efficacy, the variable names chosen to represent the 2nd and 3rd terms need to be revised, and the equals signs in equations 1, 2 and 3 replaced with proportional-to signs. Other improvements are also needed here such as to improve the variable naming and explanation of this novel and effective technique to decompose the BC aAOD into the 3 driving influences. For example "MAC_CORE" is applied to represent the absorption efficacy of the BC core, and the term "Eabs" applied to represent the effect of the coating around the BC core elevating the BC absorption (compared to pure BC core) when existing variables names with qualifying subscripts would much improve the accessibility of the approach being applied here. It is straightforward to rectify these issues however, and I have recommended in points 27 and 28 how to do that.

Response:

We think the left-hand-side in equation (1) can be recovered by multiplying the three terms on the right-hand side in equation (1) when the unit of BC burden is g m^{-2} . We clarified this point in the revised manuscript as follows: “BC burden (g m^{-2} in equations (1) and (2)) is inversely related to BC lifetime” (Line 187).

Regarding the naming of variables, we would like to use $AAOD_{550}$, MAC_{core} , and $E_{abs,coat}$ considering the reasons below. First, we found many papers using the terminology of AAOD (e.g., Bond et al., 2013) but we could not find papers using aAOD. The information of wavelength (550 nm) may be important for readers as the reviewer suggested. We therefore think $AAOD_{550}$ will be the best choice as an abbreviation of absorption aerosol optical depth at 550 nm in this manuscript. Second, in Seinfeld and Pandis, Q_{abs} is used for absorption efficiency (dimensionless), not for mass absorption cross section ($m^2 g^{-1}$). Since MAC is already used in many papers (e.g., Liu et al., 2015; Boucher et al., 2016), we would like to use MAC and MAC_{core} in this manuscript. Third, we would like to use $E_{abs,coat}$ for absorption enhancement by coating species, as the reviewer suggested (specific comment 28).

Reviewer's comment:

There are also a number of places where the text needs to be toned down, so as to be clear the BC mixing state is just one of several factors that affect BC DRE efficacy. For example the 2nd sentence "Particle size and mixing state determine the lifetime and solar absorption efficiency of BC" needs to be revised because other factors such as the nature of the precipitating clouds (e.g. ice water content) and surface roughness/vegetation also affect BC deposition processes. I understand of course that the authors are making the point that, those other things being equal, these effects play a primary role. But then the wording should be changed to express that more accurately, i.e. replace "determine" with "are both strongly influential for" or "are both primary determining factors for". That way it is clear the two effects do not determine the effect on their own (it is not "determined" by them).

Response:

This comment is related to the specific comments 2) and 6). We revised the text following these reviewer's comments as follows: "Particle size and mixing state determine the solar absorption efficiency of BC and also strongly influence how effectively BC is removed, but they have large uncertainties. Here we use a multiple-mixing-state global aerosol microphysics model and show that the sensitivity (range) of present-day BC direct radiative effect, due to current uncertainties in emission size distributions, is amplified 5-13 times ($0.18-0.42 W m^{-2}$) when the diversity in BC mixing state is sufficiently resolved." (Lines 17-23).

Reviewer's comment:

I suggest when revising the text, the lead author go through (with an English native speaker) the text and make sure that other similar statements are put into context appropriately. I have listed a number of these in my specific revisions, but the co-authors should go through and together make any additional required amendments.

Response:

We thank the reviewer for providing us with many suggestions for improving the manuscript. We have now revised the text following the reviewer's suggestions (please see the responses to the specific comments below).

Reviewer's comment:

1) Lines 20-21: The authors need to add "Post-industrial increases in" or similar to be clear the warming effect is because black carbon concentrations have increased. There would be no climate effect if they had been present at the same concentration (aside from changes in their mixing state).

Response:

We revised the sentence as the reviewer suggested: "Post-industrial increases in atmospheric black carbon (BC) have a large but uncertain warming contribution to Earth's climate." (Lines 16-17).

Reviewer's comment:

2) Lines 21-22: The word "determine" is appropriate here re: the size and mixing states relationship to solar absorption (it is controlled by it). However, the lifetime of the BC does not just depend on its size and mixing state. For example the nature of the precipitation and of the clouds from which the precipitation is generated will also affect how effectively the BC is removed. Suggest to remove the words "lifetime and" and instead put that at the end as "and are also strongly influence how effectively they are removed".

Response:

We revised the sentence as the reviewer suggested: "Particle size and mixing state determine the solar absorption efficiency of BC and also strongly influence how effectively BC is removed, but they have large uncertainties." (Lines 17-19).

Reviewer's comment:

3) Lines 22-23: this sentence is the authors opinion and is not appropriate as currently worded. I think this is the corollary the reader should take away from the paper -- but then this statement should be reserved for later in the Abstract, perhaps after the next three sentences that state the main results (at line 30). So that sentence could be reworded to follow on as: "... total aerosol DRE changes from positive (warming effect) to negative (cooling effect). We therefore contend that aerosol models should either be improved to represent such BC mixing state variations, or else have their BC DRE uncertainty range amplified to account for the effects they are omitting."

Response:

We deleted this sentence considering the limitation of the number of words for Abstract.

Reviewer's comment:

4) Lines 23-24: Suggest to add, ", by running a variable-mixing-state aerosol microphysics model," or similar between "We show" and "that present-day" (it's important, in my opinion, to mention the way that this is shown here). With that change you can remove "in a global aerosol model".

5) Line 27-28: Re-word "present-day BC DRE changes between 0.18 and 0.42 W/m²" to be clearer what you mean. Maybe just case of adding "by" between "changes" and "between"?

Response:

We revised the text considering the reviewer's two comments on this sentence as follows: "Here we use a multiple-mixing-state global aerosol microphysics model and show that the sensitivity (range) of present-day BC direct radiative effect, due to current uncertainties in emission size distributions, is amplified 5-13 times (0.18-0.42 W m⁻²) when the diversity in BC mixing state is sufficiently resolved." (Lines 19-23).

Reviewer's comment:

6) Line 42: As in my general comment above, the words "are controlling the burden" need to be revised because there are other factors too influencing these effects, so it is not just these two that are controlling. Suggest to revise "are controlling the burden (or lifetime) and heating effect..." to instead say "affect the BC removal rates and how strongly the surrounding air is heated." or similar.

Response:

We revised the text as the reviewer suggested as follows: "The enhancements of absorption efficiency and CCN activity then affect BC removal rates and how strongly the surrounding air is heated^{1,8-10}." (Lines 36-37).

Reviewer's comment:

7) Line 44: Replace "have various particle size and composition (mixing state, MS)..." with "have a range of particle sizes and mixing states (MS)..." -- I think that's clearer.

8) Line 45: Suggest to join this sentence up with the previous one, so have the previous sentence extended and revised from : "...even in the same air mass. These particles have different..." to instead say ", with the same air mass thereby having different..."

Response:

We revised and combined the two sentences as the reviewer suggested as follows: "BC particles have a wide range of particle sizes and mixing states (MS), even in the same air mass^{4,5}, thereby having different absorption efficiency and CCN activity because enhancements to absorption efficiency and CCN activity increase with increasing BC particle size and the fraction of non-BC species^{7,9} (Fig. 2 in Matsui⁹)" (Lines 38-41)

Reviewer's comment:

9) Line 48: Suggest to replace "However, these particle size and MS-dependent..." with "Such particle size and MS-dependent..." (better English), and then, again suggest to join up the current short final sentence of the para with this one, as "... sufficiently in most global aerosol models, which tend to have only 1 or 2 ..."

Response:

We revised the text as the reviewer suggested as follows: "Such particle-size- and MS-dependent absorption efficiency and CCN activity are not represented sufficiently in most global aerosol models, which tend to have only 1 or 2 BC MS

categories (e.g. hydrophobic, hydrophilic) and 3 or 4 categories for particle sizes¹¹⁻¹⁷, except for some advanced aerosol models^{18,19}.” (Lines 41-45)

Reviewer’s comment:

10) Line 53: Suggest to re-word as "changes in aerosol size distributions in emissions only" with "changes in assumed size distribution at emission, but only" (better English).

Response:

We revised the text as the reviewer suggested as follows: “Here we use a state-of-the-art global aerosol model which resolves both particle size and MS of BC^{10,24} and show that the direct radiative effect (DRE) of BC is highly sensitive to changes in assumed size distributions at emission^{21,25,26}, but only when the model resolves BC MS diversity sufficiently.” (Lines 57-60).

Reviewer’s comment:

11) Lines 54-58: Need to make this sentence clear that this is what your study is showing. Suggest to re-word the start of the sentence changing "Changes in emissions..." to "Our simulations show that differences in emissions..." Then later in the sentence re-word "produce an unrecognized large uncertainty in BC DRE due to altering three BC properties simulated by the MS-resolved model: ..." with "produce a much larger uncertainty in BC DRE when the MS variations are resolved. The hitherto unrecognized effect occurs because the MS-resolving model simulates three key properties of the BC differently when the MS variations are simulated: ..."

Response:

We revised the text as the reviewer suggested as follows: “Our simulations show that differences in emission size distributions within their current uncertainty ranges produce a much larger range of BC DRE when the MS variations are sufficiently resolved. The hitherto unrecognized effect occurs because the MS-resolving model simulates three key properties of the BC differently when the MS variations are simulated: atmospheric lifetime of BC, absorption efficiency of BC core, and absorption enhancement by coating species.” (Lines 60-66).

Reviewer’s comment:

12) Line 61: Change the start of this 1st sentence of the paragraph to say "We carry out model experiments to quantify the importance of resolving the size-resolved BC mixing state by comparing...." (delete the specifics of the "multiple MS categories (here 8 MS categories) is" as this is in the methods.

Response:

We revised the text as the reviewer suggested as follows: “We carry out model experiments to quantify the importance of resolving the size-resolved BC mixing state by comparing global simulations using a BC-MS-resolved representation (Multiple-MS) to a single BC MS representation (Single-MS) within the same model framework (see Methods and Supplementary Table 1).” (Lines 71-74).

Reviewer's comment:

13) Lines 64-67: Suggest to start this sentence from "Since all aerosol processes" with "In these model experiments, all aerosol processes..." and then replace "treatment, Multiple-MS can calculate.." with "treatment, with Multiple-MS a benchmark simulate that realistically calculates.." -- with that change of wording can delete "in detail" from the end of the sentence (it's better to state that this is done realistically than say it's done in detail).

Response:

We revised the text as the reviewer suggested as follows: "In these model experiments all aerosol processes are calculated with the particle-size- and MS-resolved treatment, with Multiple-MS the benchmark simulation that realistically calculates absorption efficiency and CCN activity of BC and their enhancements by aging processes^{24,27}." (Lines 74-78).

Reviewer's comment:

14) Lines 67-68: Re-word "In contrast, Single-MS cannot calculate these properties because all particles are assumed to be thickly-coated BC particles." with "The Single-MS run is designed to represent the simplified BC treatment used in most models, all particles assumed to have the same coating thickness for BC particles."

Response:

Considering that the coating thickness of all BC particles is the same at a single grid box at a given time, but vary three-dimensionally and with time in the Single-MS, we revised the sentence as follows: "The Single-MS run is designed to represent the simplified BC treatment used in most aerosol models: all BC containing particles, in a single grid box at a given time, are assumed to be have the same coating thickness." (Lines 78-80).

Reviewer's comment:

15) Lines 68-69: Re-word "We primarily use simulations with the Small, Base and Large emission size...." as "With each model set-up, experiments are carried out with Small, Base and Large emissions size...."

Response:

We revised the text as the reviewer suggested as follows: "With each model set-up, experiments are carried out with the Small, Base, and Large emission size distributions^{25,26} (Supplementary Table 2)." (Lines 80-82).

Reviewer's comment:

16) Lines 70-73: Explain to the reader the reason why you carry out this AeroCom analysis, by changing the start of this sentence from "We also use seven additional simulations, which are based on a range of emissions size distributions used in the Aerosol Comparisons..." with "To put our results in the context of current assumptions in models, we also carry out seven additional simulations applying the emissions sizes used by models participating in the Aerosol Comparisons..." with this change of wording you can then delete the "models, to support the conclusions from the

Small-, Base- and Large-size simulations." (just finish the sentence after citing the reference 21).

Response:

We revised the text as the reviewer suggested as follows: "To put our results in the context of current assumptions in prominent global aerosol models, we also carry out seven additional simulations applying the emission size distributions used by models participating in the Aerosol Comparisons between Observations and Models (AeroCom) project²¹." (Lines 82-85).

Reviewer's comment:

17) Lines 75-77: Need to change the wording of this sentence – suggest to re-word as "Our results demonstrate the importance of accurately identifying the emissions size distribution, but additional simulations (see Supplementary Discussion S1) show the results are not sensitive to the exact mixing state assumed at emission."

Response:

We deleted this sentence from Introduction following the specific comment 19). We used the latter part of the suggested sentence in Results section as follows: "These results show that our main conclusions are insensitive to the exact BC mixing state assumed at emission." (Lines 171-172).

Reviewer's comment:

18) Lines 79-83: Move the citing of references 22 and 23 to be earlier in the sentence, to come just after "Our previous studies" (and delete the "also").

Response:

We revised the text as the reviewer suggested as follows: "Our previous studies^{27,28} showed that the MS-resolved representation could reproduce important observed features of BC MS, such as number fractions of BC-containing and BC-free particles and coating amounts of BC-containing particles with their size dependences, which are neglected in the single MS representation." (Lines 92-95).

Reviewer's comment:

19) Lines 83-87: This is too much detail at this stage -- this text and that in lines 75-83 -- needs to come after the main results have been presented, in a Discussion section of the manuscript. Please go through the manuscript and think through the sequencing of the text -- the Introduction certainly needs a couple of additional sentences at the start to motivate the importance of the study with reference to other previous related studies. And moving this section to later could accommodate that addition whilst keeping the Introduction the same length.

Response:

We moved some sentences to the latter part of this manuscript as the reviewer suggested. The following sentence was moved to the paragraph where the MS assumption in emissions is discussed: "These results show that our main conclusions are insensitive to the exact BC mixing state assumed at emission." (Lines 171-172).

The following sentences were also moved to the paragraph where the MS treatment for internally-mixed BC particles is described: “The core-shell MS treatment (BC makes the core while other species make the shell) is used for internally-mixed BC particles in our simulations²⁴.” (Lines 237-238) and “The conclusions obtained in this study do not change even if we use different MS treatments for internally-mixed BC particles (see Methods and Supplementary Fig. 8).” (Lines 242-244).

We also added a paragraph to Introduction to clarify the motivation of this study and the results of previous studies as the reviewer suggested: “There are large uncertainties in the treatment of aerosol size distributions at emission in current global aerosol models^{20,21} and Reddington et al.²⁰ has shown the importance of reducing these uncertainties, especially when estimating aerosol number concentrations – a key parameter in accurately estimating aerosol-radiation and aerosol-cloud interactions. Previous studies^{17,22,23} have also shown the impact of altering assumed emission size distributions on atmospheric BC mass concentrations, but their importance remains unclear. For example, Reddington et al.²² and He et al.¹⁷ suggested negligible importance of emission size distributions to BC mass, while Bauer et al.²³ showed high sensitivity of BC mass to emission size distributions. In addition, the impact of assumptions about emission size on BC radiative effects is not discussed in the literature.” (Lines 46-56).

Reviewer’s comment:

20) Line 91 -- Add an introductory sentence to first identify/establish the rationale behind the analysis you are carrying out -- to assess the importance for the radiative effects of BC of resolving its mixing state through each of the three aspects identified earlier.

Response:

Considering the reviewer’s comment, we added a few sentences to the beginning of this subsection (Sensitivity of black carbon direct radiative effect) before describing the difference between Multiple-MS and Single-MS as follows: “The Small-size simulations have a longer BC lifetime and higher BC burden than the Large-size simulations for both the Multiple-MS and Single-MS representations because larger particles generally have higher CCN activity (lower critical supersaturation required to form a cloud droplet), which translates into a shorter BC lifetime since wet removal is the dominant BC sink^{24,29}. Simulated BC lifetimes in both Multiple-MS (4.5-6.0d) and Single-MS (4.3-5.1d) tend to be shorter than the AeroCom multi-model average (6.5 d)³⁰, which is generally consistent with previous studies^{30,31} suggesting a too long BC lifetime in many AeroCom models. However, the overestimation of upper tropospheric BC in many global aerosol models^{30,32} is also seen in the Multiple-MS and Single-MS simulations shown here (Supplementary Fig. 1). To examine the sensitivity of BC burden and its radiative effect to assumed size distributions in emissions, we use the ratio of global-mean BC values between the Small-size and Large-size simulations (Small/Large ratio).” (Lines 99-111).

Reviewer’s comment:

21) Line 92,93 -- the fact that the single-MS representation has only 19% variation compared to 32% resolved in the multiple-MS case is important and

could be re-iterated/re-interpreted in the Abstract as the single-MS missing 68% (32/19-1) of the variability from the different size distribution (you could cite an example of this arising from different types of combustion sources for example). This is only for the BC lifetime, and it could be combined with the others to provide a summary in the Abstract to give a quantitative measure to report objectively to allow the reader to determine how important it is for models to resolve the effect.

Response:

We thank the reviewer for suggesting this important analysis. We compared the variability range of BC lifetime, MAC_{core} , and $E_{abs,coat}$ between the Multiple-MS and Single-MS representations and added the following sentences to the Results section: “The variability range of BC lifetime is underestimated by 44% in the Single-MS representation (4.5-6.0d for Multiple-MS and 4.3-5.1d for Single-MS (Fig. 1a))” (Lines 191-193), “The range of MAC_{core} is underestimated by 72% in the Single-MS representation (4.2-5.9 $m^2 g^{-1}$ for Multiple-MS and 5.4-5.9 $m^2 g^{-1}$ for Single-MS (Fig. 5a)).” (Lines 203-205), and “The variability range of $E_{abs,coat}$ is underestimated by 63% in the Single-MS representation (1.5-1.8 for Multiple-MS and 2.2-2.3 for Single-MS (Supplementary Table 3)).” (Lines 221-223).

We also added these results to the Abstract as follows: “This amplification is caused by the lifetime, core absorption, and absorption enhancement effects of BC, whose variability is underestimated by 45-70% in a single-mixing-state representation.” (Lines 23-25).

Reviewer’s comment:

22) Lines 93-96 -- The authors need to qualify their assessment however with some reference to the fact that many of the single-MS models will likely assign the particles/mode/size-bin properties that implicitly provide a hygroscopicity and CCN-activity that will likely represent a low-coated BC particle rather than a pure BC particle (even if that is what the model actually resolves).

Response:

We added the following sentences to the main text considering the reviewer’s comment: “The sensitivity of typical global aerosol models is likely to be somewhat higher, although there is the potential for smaller sensitivity, than that in Single-MS depending on the representation of the properties of aerosol particles (e.g., number of MS and size bins/modes) within the model.” (Lines 118-122).

Reviewer’s comment:

23) Line 95 -- it is likely not to be obvious to some readers that a larger variability in CCN-activity necessarily results in a larger variation in BC lifetime. Perhaps it's just a case of removing "This is because Multiple-MS resolves..." to instead say "This stronger BC lifetime sensitivity results from Multiple-MS resolving..." (and also add a comma after "thinly coated BC"), and additionally adding a sentence after that to explain the link to BC lifetime, suggest something like: "In the CAM5-chem/ATRAS model, higher CCN activity translates into a shorter BC lifetime since wet removal is its dominant sink (e.g. Textor et al., 2006; Matsui, 2017)."

Response:

We revised the text as the reviewer suggested: “This stronger BC lifetime sensitivity results from Multiple-MS resolving pure BC and thinly-coated BC” (Lines 114-115). The 2nd suggested sentence was used in the first sentence of this subsection as follows: “The Small-size simulations have a longer BC lifetime and higher BC burden than the Large-size simulations for both the Multiple-MS and Single-MS representations because larger particles generally have higher CCN activity (lower critical supersaturation required to form a cloud droplet), which translates into a shorter BC lifetime since wet removal is the dominant BC sink^{24,29}.” (Lines 99-103).

Reviewer’s comment:

24) Lines 93-96 -- Has this sensitivity to BC emissions size distribution been assessed before? Can these numbers for the emissions-size sensitivity for BC lifetime be compared to those derived from other studies? For example Reddington et al. (2013), using a sectional global aerosol microphysics model with 3 distributions of bins that contain BC, present results which illustrate similar sensitivity of simulated surface BC to assumptions on emissions size distribution, finding higher surface BC for larger assumed size than smaller size (e.g. as 76 and 122 ng/m³ on the EUCAARI campaign mean). The model applied in that analysis did not resolve the effect of the mixing state on "CCN activity", but did resolve variations in the rate of "condensation-ageing" across the particle size range, and identified associated slower ageing as the reasons for the higher BC in the small emissions size case. The authors need to add some reference to this work either here or in some discussion prior to the conclusions of the paper. The analysis in Reddington et al. (2013) also compared the BC coating thickness (across the size range) to that measured by the SP2 instrument, and the authors should refer to how this compares also in their model, or if they have evaluated it in their papers, explain how (in broad terms) their simulated size resolved BC coating thickness compares to that shown in that paper for the simulations there with that particular 3-mixing-state sectional approach. Suggest to add 1 sentence to this paragraph referring to these results.

Response:

The number fraction of BC-containing and BC-free particles in Reddington et al. (2013) (Table 6 in their paper) is very interesting and useful. We added this result to the main text as follows: “The number fraction of BC-free particles to the total is 70-80% in East Asia, 75-90% in Europe, and 90% over the Arctic in both measurements^{22,27,39} and Multiple-MS, whereas BC-free particles are not treated (0%) in Single-MS.” (Lines 208-210).

Regarding the coating thickness of BC in Reddington et al. (2013), they described that observed coating thickness was not available in their study (they showed simulated coating thickness only). BC coating thickness was evaluated by SP2 measurements in East Asia in our previous studies, as described at Lines 90-95.

Additional to Reddington et al., two further studies assessing the sensitivity of BC burden to emission size distributions (Bauer et al., 2010; He et al., 2016) are added to the manuscript in the Introduction, as described in the response to the specific comment 19) (Lines 46-56).

In Reddington et al. (2013), the response of BC mass to emission sizes looks

opposite (larger emission size has larger BC mass), but this is seen only for BC within the size range of their SP2 instrument (McMeeking et al., 2010). Since their conclusion was that total BC mass is not sensitive to emission particle sizes, we added this description to Introduction. It is difficult to evaluate the causes of different sensitivity between Reddington et al. (2013) and Bauer et al. (2010) because their model representations and setups are totally different.

We also added the following sentence to the main text: "This tendency of BC lifetime to be longer and BC burden to be larger with smaller emission size is qualitatively consistent with Bauer et al.²³." (Lines 117-118).

Reviewer's comment:

25) Lines 97-105 -- The acronym AAOD is used in the paper to denote absorption aerosol optical depth, but I recommend to change this "AAOD" to "aAOD" since the AOD is an established three-letter-acronym (TLA), the absorption then being a qualifying term that should be in lower case ahead of that established TLA. I would also recommend the authors always provide the wavelength in subscript whenever the acronym aAOD or AOD is used in papers. So change "AAOD" to "aAOD" always with subscript 550 if this is the only wavelength assessed.

Response:

As described in the response to the fifth major comment, we would like to use AAOD₅₅₀ in this manuscript.

Reviewer's comment:

26) Line 116 -- replace "Figs 1e-1f" with "Figs 1e-f".

Response:

As described in the response to the third major comment, we deleted Figure 1f from the manuscript. We changed this part to Fig. 3.

Reviewer's comment:

27) Lines 121-127 -- the decomposition of the BC aAOD₅₅₀ into the three driving influences is a novel and effective way to explain why the single-MS approach is under-representing the uncertainty in BC DRE. However, the equations 1, 2 and 3 on lines 122-124 are not really correct, as each one is currently stating that the numerical value of the term on the left-hand-side can be recovered simply by multiplying the terms on the right-hand side. I think the authors are really saying that the terms on the left-hand side are proportional to the product/ratio of terms shown on the right-hand side, in which case the problem is easily remedied by replacing the equals sign instead with a "proportional-to" sign. Also, please remove "(BC + nonBC)" from the left-hand side here as the term "BC AAOD" (or BC aAOD₅₅₀ as needs to be written) is already being clear it is a "speciated AOD" (just the component of the absorption-AOD that occurs because of the BC component).

Response:

We think the left-hand-side in equation (1) can be recovered by multiplying the three terms on the right-hand side in equation (1) when the unit of BC burden is $g\ m^{-2}$.

We deleted “(BC + nonBC)” from equation (1) as the reviewer suggested and modified the explanation of BC AAOD in equations (2) and (3) as “BC AAOD₅₅₀ (BC core)” and “BC AAOD₅₅₀ (BC core + coating)”. We think this explanation is useful for readers.

Reviewer’s comment:

28) The variable names for the terms MAC_{core} and E_{abs} in equations 1, 2 and 3 also need to be improved as the paper is revised. For the absorption efficacy term, the authors should follow the variable naming from Seinfeld and Pandis, "Atmospheric Chemistry and Physics: From Air Pollution to Climate Change.....", which has the variable "Q" subscript "abs" used for the mass absorption cross-section. The authors could specify this as "Q" subscript "abs,core" to be clear it is applied to the BC core in this case. For the E_{abs} term, really, this subscript "abs" is not quite the correct qualifying term (on its own at least), as the effect is really, as explained on page 6, that the coating effect promotes absorption over that which would be the case for a pure BC particle with the same amount of black carbon mass. I therefore recommend to change the name of the term "E" subscript "abs" to instead be with "E" subscript "coat" or else subscript "abs,coat" or "abs,core-shell". That way it will be immediately clear to the reader that that term is representing the enhancement of the absorption due to the coating or core-shell effect.

In addition to the above naming changes, I suggest an overline or hat symbol ("^") is added to each term because these terms are not providing any specific gridbox value, but are rather an average or central value, representative of how the effect manifests in the column-integrated BC-attributed absorption (the BC absorption optical depth).

Response:

We would like to use MAC_{core} and E_{abs,coat} in this manuscript as described in the reply to the fifth major comment.

Since almost all values in this manuscript are global-mean values, we added the following sentence to the text: “Unless noted otherwise, global- and 5-year mean values are shown in this study.” (Lines 95-96). We also added overlines to the variables in equations 1, 2, and 3, as the reviewer suggested.

Reviewer’s comment:

29) Lines 127 to 130 -- the sentence beginning on line 127 needs to be moved to the start of this section (i.e. begin the paragraph with this), the text after "The three terms in the right side of the equation" deleted and the remainder of the sentence then following after "to three effects" in the current 1st sentence, also with an initial brief few words to explain why this is being done such as "To explain why single-MS approaches are under-representing BC DRE uncertainty, we decompose BC aAOD₅₅₀ into three component elements:" then add the text from the 2nd half of the current lines 127-130.

Response:

We revised the text as the reviewer suggested as follows: “To explain why Single-MS is under-representing the BC DRE range, we decompose BC AAOD₅₅₀ into three effects: 1) the BC lifetime effect, 2) absorption efficiency of BC core (mass

absorption cross section of BC core (MAC_{core}), and 3) absorption enhancement by coating species ($E_{abs,coat}$).” (Lines 175-178).

Reviewer’s comment:

30) Line 131 -- "...since the mass flux of emissions is held constant." Suggest to reword "is held constant" to "is the same in all simulations". Also, need to explain that the change in size results in much more particles being emitted, suggest to add "the number of particles emitted increasing as emission size decreases" or similar to briefly remind the reader of this.

Response:

We revised the text as the reviewer suggested as follows: “BC burden ($g\ m^{-2}$ in equations (1) and (2)) is inversely related to BC lifetime, since the mass flux of emissions is the same in all simulations.” (Lines 187-188). The description of "the number of particles emitted increasing as emission size decreases" was added to the first paragraph of the Results section because this information is important: “The mass flux of aerosol emissions is the same in all simulations (Supplementary Table 3), with the number of particles emitted increasing as the median aerosol size at emission decreases.” (Lines 87-89).

Reviewer’s comment:

30) Line 133-134 -- The authors have a 1-sentence paragraph here which looks like somehow how the submitted manuscript is missing some key text. Suggest to add 2 or 3 sentences to complete this paragraph providing explanation of how the emissions-size influences the BC burden/lifetime in the CAMS5-chem/ATRAS model and how other models do this differently. This is one of the key points that provide the explanation for why the single-MS approach under-represents the BC burden/lifetime sensitivity to emissions-size.

Response:

We added the following few sentences to the main text, considering the reviewer’s comment: “The variability range of BC lifetime is underestimated by 44% in the Single-MS representation (4.5-6.0d for Multiple-MS and 4.3-5.1d for Single-MS (Fig. 1a)) because explicitly calculating the size- and MS-dependent critical supersaturation of BC-containing particles in the Multiple-MS covers a broader range of CCN activity (including the less CCN-active population of particles such as pure BC and thinly-coated BC). In Single-MS only a size-dependent critical supersaturation is calculated so all BC-containing particles are more CCN-active than the pure BC and thinly-coated BC resolved in Multiple-MS.” (Lines 191-198).

Reviewer’s comment:

31) Figure 1 -- the authors have explained in the text (lines 360-362) that the uncertainty ranges presented here are not exactly the same as in the 5 to 95% range given in the Bond et al. (2013) BC assessment. Although that is true, if a probability could be given for the limit values in particle size considered (e.g. the upper size limit is 90% or 95%) then the ranges given could be converted into a comparable confidence interval. Could the authors give some statement about their expert judgement for what

the small-size and large-size represent? Can this be considered probabilistically? Perhaps it would not be appropriate to convert into the range because the authors are really testing the sensitivity (so it's a sensitivity not an uncertainty). If that is the case then the authors should add a sentence explaining why the range given cannot be similarly converted into an uncertainty range to be compared on a similar basis.

Response:

We deleted Figure 1f following a comment by Reviewer #3. Please see the response to the third major comment.

=== (the response to the third major comment)

We deleted comparisons with the range in Bond et al. BC assessment (Figure 1f), considering a comment by Reviewer #3. This deletion does not change our main conclusions because we mainly use Figure 1e (comparisons with the AeroCom models; Figure 3 in the revised manuscript) for comparisons with the ranges by other aerosol models.

As described in Lee et al. (2013), carbonaceous aerosol particle size is a very uncertain parameter because the size of combustion particles from fossil fuels depends on the details of the source (e.g. fuel and industry type and age of the facility) and the size from biomass burning and wildfires depends on the burning efficiency of the fuel and its properties (e.g., moisture content and fuel type). Their ranges of emission particle sizes correspond to the maximum-minimum ranges of median diameters of emissions assumed in the AeroCom models (Supplementary Table 2). The range of median diameters for fossil fuel sources is 30-80 nm in both Lee et al. (2013) and the AeroCom models. The range for biofuel and biomass burning sources in Lee et al. (2013) (50-200 nm) are slightly larger the AeroCom models (30-150 nm), but a recent study suggests carbonaceous biomass burning emission could have 300 nm in diameter in maximum (Regaryre et al., 2018).

We added these sentences to Methods (Lines 436-446) in the revised manuscript.

===

Reviewer's comment:

32) Figure 2 -- I really like this diagram as the illustration really helps the reader understand visually what is being explained in the text. However, I don't think the idea to have the blue and red arrows showing positive and negative responses really works (at least not as currently shown). Why are the arrows to the right of the boxes in red and the arrows to the left in blue? And why are there "(+/-)" signs all in red above each box. Also, the left-hand box says "Particle size (emission, atmosphere)" but that text would be better understood by the reader if it was changed to "Emissions particle size". Certainly it is not necessary to state "atmosphere". Also, it is my opinion that this Figure is fine to just illustrate the 3 driving influences/components of the BC aAOD550 that the authors decomposed in equations 1, 2 and 3. With this in mind, I suggest that, when revising the paper, the authors simplify this Figure, having just one arrow in each of the three drivers/components that are connecting up from the emissions particle size on the left to the aAOD550 on the right.

Response:

We simplified Figure 2 considering the reviewer's comment. Please see the response to the second major comment.

==== (the response to the second major comment)

We simplified Figure 2 (Figure 4 in the revised manuscript) by removing (+/-) signs and the description of "pure-BC, thinly-coated BC", "BC-free", and "Thickly-coated BC" and by changing "Particle size (emission, atmosphere)" to "Emissions particle size" (as the reviewer suggested). We added the explanation of blue and red arrows to the figure caption to clarify their meanings: "Specifically, decreasing particle sizes at emission (e.g. Small-size simulation) increases BC lifetime, BC core absorption, and absorption enhancement, and these enhancements increase absorption aerosol optical depth at 550 nm (AAOD₅₅₀) and DRE of BC in the Multiple-mixing-state representation." (Lines 807-810). We want to retain the information of DRE and their values because understanding BC DRE responses is the main purpose of these analyses.

====

Reviewer's comment:

33) Figure 3 -- replace "MAC_{core}" with Q-subscript-abs,core as in comment 29. Also, on line 674 the text "theoretical calculations" needs to be better qualified with the mention of the method used to translate the different mixing state into absorption cross-section.

Response:

As described in the response to the fifth major comment, we would like to use MAC_{core} in this manuscript. We clarified the method of theoretical calculations by adding "(using the algorithms of Bohren and Huffman based on the Mie theory)" to the caption of Figure 3 (Figure 5 in the revised manuscript) (Lines 821-822).

References:

Bauer, S. E., Menon, S., Koch, D., Bond, T. C. & Tsigaridis, K. A global modeling study on carbonaceous aerosol microphysical characteristics and radiative effects. *Atmos. Chem. Phys.* **10**, 7439-7456 (2010).

Bond, T. C. *et al.* Bounding the role of black carbon in the climate system: A scientific assessment. *J. Geophys. Res. Atmos.* **118**, 5380-5552 (2013).

Boucher, O. *et al.* Jury is still out on the radiative forcing by black carbon. *Proc. Natl. Acad. Sci. USA* **113**, E5092-E5093 (2016).

He, C. *et al.* Microphysics-based black carbon aging in a global CTM: constraints from HIPPO observations and implications for black carbon budget. *Atmos. Chem. Phys.* **16**, 3077-3098 (2016).

Lee, L. A. *et al.* The magnitude and causes of uncertainty in global model simulations of cloud condensation nuclei. *Atmos. Chem. Phys.* **13**, 8879-8914 (2013).

Liu, S. *et al.* Enhanced light absorption by mixed source black and brown carbon particles in UK winter, *Nat. Commun.* **6**, 8435 (2015).

McMeeking *et al.* Black carbon measurements in the boundary layer over western and northern Europe. *Atmos. Chem. Phys.* **10**, 9393-9414 (2010).

Regayre, L. *et al.* Aerosol and physical atmosphere model parameters are both important sources of uncertainty in aerosol ERF. *Atmos. Chem. Phys. Discuss.* <https://doi.org/10.5194/acp-2018-175> (2018).

Reddington, C. L. *et al.* The mass and number size distributions of black carbon aerosol over Europe. *Atmos. Chem. Phys.* **13**, 4917-4939 (2013).

Response to reviewer #2

NCOMMS-17-30621-T: “Impact of particle size and mixing state diversity on estimates of black carbon mitigation” by H. Matsui et al.

We thank the reviewer very much for reading the paper carefully and giving us valuable comments. We revised the paper by taking into account the reviewer’s comments. Detailed responses to individual comments and suggestions are given below.

Our important findings are the following three points. First, we showed interactions between emission size distributions and BC MS diversity made an unrecognized large BC DRE uncertainty, which is greater than the AeroCom multi-model range, through the comparisons between MS-resolved and MS-unresolved global aerosol models (Fig. 3). Second, we clarified the three causes of the 5-13 times higher sensitivity of BC DRE to emission sizes in the MS-resolved model and demonstrated the problems of typical global aerosol models (Figs 4 and 5). Third, we found the changes in emission size distributions from the present-day to the future would be important in the evaluation of BC mitigation policy (Fig. 6). In our understanding, none of these conclusions have been found in the previous studies raised by the reviewer.

Reviewer’s comment:

L 37: BC has the largest human made impact on climate change. Most of the BC in the atmosphere is from biomass burning, thus a large portion is natural. Overall I would say sulfate has the largest man made impact on climate when looking at aerosol species.

Response:

We don’t think that most of the BC in the atmosphere is from biomass burning emissions. Some studies estimated that BC emissions from biomass burning sources accounted for 30-40% of total BC emissions (Bond et al., 2013; Liu et al., 2012). 60-70% of BC emissions are from anthropogenic (fossil fuel and biofuel) sources. In addition, a significant portion of biomass burning emissions is anthropogenic (e.g. agriculture burning, tropical deforestation).

In this sentence, we focus on the forcing uncertainty. In our understanding, the uncertainty in BC forcing is much greater than that in sulfate forcing (IPCC AR5 chapter 8). Figure 8.17 in the IPCC AR5 report shows that BC is “one of the largest uncertainties” in the estimation of anthropogenic radiative forcing.

IPCC AR5 Fig. 8.17.

Reviewer's comment:

L 49 The paper claims that BC mixing state effects are not represented in global climate models. This claim is not true as all models that include aerosol microphysical schemes represent this process in some degree. Some models do even simulate those processes in great detail, such as Jacobson 2001 and Bauer et al 2008.

Response:

While these two models are already described in the Methods section (Line 419 in the revised manuscript), we agree they are not described clearly in the main text as the reviewer suggested. We revised the sentence as follows considering the reviewer's comment: "Such particle-size- and MS-dependent absorption efficiency and CCN activity are not represented sufficiently in most global aerosol models, which tend to have only 1 or 2 BC MS categories (e.g. hydrophobic, hydrophilic) and 3 or 4 categories for particle sizes¹¹⁻¹⁷, except for some advanced aerosol models (Jacobson, 2001; Bauer et al., 2008)." (Lines 41-45).

Most global aerosol models have some aerosol microphysical schemes as the reviewer pointed out, but their representations of BC mixing state are oversimplified and "not sufficient" as described in this sentence and other parts in this manuscript (described in Introduction of Matsui (2017) also).

We want to note that in Jacobson (2001) and Bauer et al. (2008) 3 BC MS

categories are defined based on the fraction of BC in a particle. In our model, 7 BC MS categories are defined by BC fractions (Methods). This greater number of BC MS categories is a key required component to obtain our important conclusions such as the large sensitivity of the BC core absorption effect, as shown by additional offline optical calculations assuming 2 and 3 MS (Lines 256-277): “Since many global aerosol microphysics models consider 2 MS categories (hydrophobic and hydrophilic), we conducted offline optical calculations, assuming 2 and 3 MS categories, using the results of Multiple-MS simulations to further highlight the importance of representing multiple BC MS. In the 2-MS treatment, we consider hydrophobic and hydrophilic BC groups (BC mass fractions are assumed as 0.9-1.0 and 0-0.9, respectively) from the 8 MS information in Multiple-MS. In the 3-MS treatment, hydrophobic BC, hydrophilic BC, and BC-free groups (BC mass fractions are assumed as 0.9-1.0, 0.0001-0.9, and 0-0.0001, respectively) are considered. The Small/Large ratios in the 2-MS treatment are 1.04 and 0.94 for the BC core absorption and absorption enhancement effects, respectively, suggesting that the 2-MS treatment cannot capture the sensitivity of these two effects as simulated by Multiple-MS (Small/Large ratios are 1.41 and 1.15, respectively, in Multiple-MS). The Small/Large ratios in the 3-MS treatment are 1.24 and 1.19 for the two effects, showing a much increased sensitivity for the two effects by the addition of resolving BC-free particles. However, the sensitivity for the BC core absorption effect is still underestimated (by 41%) compared with the Multiple-MS treatment. An important component of most global aerosol models with 2 BC MS categories is the chosen threshold value to distinguish between hydrophobic and hydrophilic BC. This aging threshold term is highly variable, and often modified in order to better correlate simulated BC properties with observations. Therefore, resolving multiple BC MS categories removes the dependence of choosing a threshold and represents the diversity of CCN activity and absorption enhancement of BC-containing particles more realistically.”

Reviewer’s comment:

L83: It has been clearly demonstrated that core-shell representation overestimate BC absorption enhancement. The results presented in the supplementary material disagree with Adachi et al (2010) and Cappa et al 2012.

Response:

In our simulations, the core-shell representation does not have higher BC absorption than other mixing-rule assumptions (DEMA and Bruggeman), as shown in Supplementary Fig. 8 and Supplementary Table 3. Absorption enhancement in our model is consistent with a recent global model simulation (Fierce et al., 2016) which used the results of a particle-resolved box model.

Some field measurements reported low absorption enhancement as the reviewer pointed out. Previous studies (e.g. Jacobson (2012) and Fierce et al. (2016)) showed that the enhancement of BC absorption by aerosol-phase water is important and may be a major reason to explain the different absorption enhancement between measurements and model simulations.

Considering these points, we added the following sentences to the manuscript:

“Global mean $E_{\text{abs,coat}}$ in the Base simulation (1.6) is similar to that in a recent global model simulation (1.5) which used the results of a particle-resolved box model⁴⁰.”

(Lines 232-234).

“The core-shell MS treatment (BC makes the core while other species make the shell) is used for internally-mixed BC particles in our simulations²⁴. This treatment is likely to be realistic because limited particle-resolved measurements have shown that attached-type BC particles (BC is at the surface of the particle), which have lower enhancement of absorption than coated-type BC particles (BC is inside of the particle)⁴¹, are very limited (< 3%) in the atmosphere^{42,43}. The conclusions obtained in this study do not change even if we use different MS treatments for internally-mixed BC particles (see Methods and Supplementary Fig. 8). Some field measurements have reported low $E_{\text{abs,coat}}$ values^{44,45}, however they are not able to observe absorption enhancement in humid regions of the atmosphere as ambient particles are usually dried (relative humidity of < 50%) in field measurements⁴⁰. Some laboratory experiments⁴⁶ and global modeling studies^{40,47} have shown the importance of BC absorption enhanced by aerosol water.” (Lines 237-248).

Reviewer’s comment:

SI: The paper compares the simulation to global mean budget numbers of other aerocom models, but is completely lacking any comparisons to observations. Global mean numbers are not a good measure to investigate locally and regional acting aerosol effects.

Response:

Extensive model comparisons with measurements were undertaken in our previous studies (Matsui and Mahowald, 2017; Matsui et al., 2013, 2014), see Lines 90-95. We have also now added comparisons of simulated MAC of BC with measurements to the revised manuscript as follows: “Mass absorption cross section (MAC) has large spatial and temporal variations: typically between 7 and 13 $\text{m}^2 \text{g}^{-1}$ at 550 nm in measurements and an average of $\sim 8 \text{ m}^2 \text{g}^{-1}$ at 550 nm in AeroCom models^{1,33-37}. Global-mean MAC is 8.8 and 13.6 $\text{m}^2 \text{g}^{-1}$ in Multiple-MS and Single-MS (Base-size simulations), respectively (Supplementary Table 3), showing that MAC in Multiple-MS is within the typical observed range while MAC in Single-MS is above the typical observed range.” (Lines 130-135).

We want to retain the focus on the global-mean aspects in this manuscript to highlight our conclusions obtained in this study. However, regional and local aspects will also be important, as the reviewer pointed out, so we have moved a short paragraph discussing sub-global spatial distributions of the sensitivity of BC DRE to the main text and discussed them using Fig. 2 as follows: “The Small/Large ratio of DRE in Multiple-MS is higher than that in Single-MS all over the globe (Fig. 2 and Supplementary Figs 3 and 4). Multiple-MS and Single-MS have similar spatial patterns: Small/Large ratio is lower near source regions at low and mid-latitudes and higher over remote regions and at high-latitudes (Fig. 2). These results show the sensitivity of BC DRE to emission size distributions is enhanced by the aging processes during transport, especially when MS is sufficiently resolved. The contrast of BC DRE between the Northern and Southern Hemispheres is a factor of 2 different between the Small-size and Large-size simulations in Multiple-MS (Fig. 1d). In Single-MS, this contrast is not sensitive to emission size distributions. A multiple MS-resolved treatment is therefore important for accurate calculations of the contrast of BC DRE

between the Northern and Southern Hemispheres and between land and ocean (Figs 1d and 2 and Supplementary Fig. 4).” (Lines 143-154).

Reviewer’s comment:

L117: Most or all climate models use BC enhancement factors in order to parameterize BC mixing state effects. Even so the process is not physically resolved in all models, it is not absent in the overall results.

Response:

Some models assume a constant absorption enhancement factor or calculate absorption enhancement based on bulk (total) mass concentrations of BC and non-BC species. However, these treatments cannot calculate the high sensitivity of BC DRE to emission size distributions in Multiple-MS because particle size and MS of BC are important for BC DRE estimates as shown in this study. We clarified this point in the text as follows: “Our results show that altering microphysical processes changes BC DRE significantly even when similar total (bulk) mass concentrations of BC and non-BC are simulated.” (Lines 325-327).

Reviewer’s comment:

The impact of BC emission size on BC aerosol forcing have been investigated in Bauer et al (2010).

Response:

Bauer et al. (2010) discussed the response of BC mass and total aerosol direct and indirect effects to emission sizes but didn’t discuss BC DRE directly. They discussed the impact of emission sizes on total aerosol effects, while our study focuses on BC DRE. We think Bauer et al. (2010) is a very good study, but none of our three main conclusions, described at the beginning of this response, are discussed in their paper.

We cited Bauer et al. (2010) in Introduction as a previous study focused on emission sizes (Lines 46-56).

Reviewer’s comment:

L222 The impact of biofuel vs fossil fuel emissions have been discussed in Jacobson (multiple papers see below) and Chen et al (2010)

Response:

Of the five Jacobson’s papers the reviewer raised, we think only Jacobson (2010) discusses the effects of reduced biofuel and fossil fuel emissions. We added this paper and Chen et al. (2010) as previous studies which discussed the impact of biofuel and fossil fuel emissions on BC mitigation (Line 340). From our understanding, they didn’t discuss the sensitivity of BC DRE or mitigation to emission size distributions which we focused on in this study.

Reviewer’s comment:

A model description Page 12 to 20 should not be published in a Nature paper.

Response:

We retained the Method section in the text following the Editor's suggestion.

Reviewer's comment:

My overall comment is that this is a very nice and interesting study. But not suitable for the journal submitted here, due to the lack of new and original material. The paper lacks key publications on the topic of BC mixing state and thus its findings are not completely unique. I would recommend to include a data evaluation part in this paper and resubmit it to a different journal, such as ACP or JGR.

Response:

We have now included model comparisons with aircraft and MAC measurements in the revised manuscript. Other extensive evaluations were undertaken in our previous studies (Matsui and Mahowald, 2017; Matsui et al., 2013, 2014).

As described in the response to individual comments, none of the papers the reviewers raised have discussed our three important conclusions: First, we showed interactions between emission size distributions and BC MS diversity made an unrecognized large BC DRE uncertainty, which is greater than the AeroCom multi-model range, through the comparisons between MS-resolved and MS-unresolved global aerosol models (Fig. 3). Second, we clarified the three causes of the 5-13 times higher sensitivity of BC DRE to emission sizes in the MS-resolved model and demonstrated the problems of typical global aerosol models (Figs 4 and 5). Third, we found the changes in emission size distributions from the present-day to the future would be important in the evaluation of BC mitigation policy (Fig. 6).

These conclusions, which have not been obtained by previous studies, contain a lot of new and original material.

References:

Bauer, S. E. *et al.* MATRIX (Multiconfiguration Aerosol TRacker of mIXing state): An aerosol microphysical module for global atmospheric models. *Atmos. Chem. Phys.* **8**, 6003-6035 (2008).

Bauer, S. E., Menon, S., Koch, D., Bond, T. C. & Tsigaridis, K. A global modeling study on carbonaceous aerosol microphysical characteristics and radiative effects. *Atmos. Chem. Phys.* **10**, 7439-7456 (2010).

Bond, T. C. *et al.* Bounding the role of black carbon in the climate system: A scientific assessment. *J. Geophys. Res. Atmos.* **118**, 5380-5552 (2013).

Chen, W.-T., Lee, Y. H., Adams, P. J., Nenes, A. & Seinfeld, J. H. Will black carbon mitigation dampen aerosol indirect forcing? *Geophys. Res. Lett.* **37**, L09801 (2010).

Fierce, L., Bond, T. C., Bauer, S. E., Mena, F. & Riemer N. Black carbon absorption at the global scale is affected by particle-scale diversity in composition. *Nat. Commun.* **7**,

12361 (2016).

Jacobson, M. Z. Strong radiative heating due to the mixing state of black carbon in atmospheric aerosols. *Nature* **409**, 695-697 (2001).

Jacobson, M. Z. Short-term effects of controlling fossil-fuel soot, biofuel soot and gases, and methane on climate, Arctic ice, and air pollution health. *J. Geophys. Res.* **115**, D14209 (2010).

Jacobson, M. Z. Investigating cloud absorption effects: Global absorption properties of black carbon, tar balls, and soil dust in clouds and aerosols. *J. Geophys. Res.* **117**, D06205 (2012).

Liu, X. *et al.* Toward a minimal representation of aerosols in climate models: description and evaluation in the Community Atmospheric Model CAM5. *Geosci. Model Dev.* **5**, 709-739 (2012).

Matsui, H. Development of a global aerosol model using a two-dimensional sectional method: 1. Model design. *J. Adv. Model. Earth Syst.* **9**, 1921-1947 (2017).

Matsui, H. & Mahowald, N. Development of a global aerosol model using a two-dimensional sectional method: 2. Evaluation and sensitivity simulations. *J. Adv. Model. Earth Syst.* **9**, 1887-1920 (2017).

Matsui, H. *et al.* Development and validation of a black carbon mixing state resolved three-dimensional model: Aging processes and radiative impact. *J. Geophys. Res. Atmos.* **118**, 2304-2326 (2013).

Matsui, H., Koike, M., Kondo, Y., Fast, J. D. & Takigawa, M. Development of an aerosol microphysical module: Aerosol Two-dimensional bin module for foRmation and Aging Simulation (ATRAS). *Atmos. Chem. Phys.* **14**, 10315-10331 (2014)

Response to reviewer #3

NCOMMS-17-30621-T: “Impact of particle size and mixing state diversity on estimates of black carbon mitigation” by H. Matsui et al.

We thank the reviewer very much for reading the paper carefully and giving us valuable comments. We revised the paper by taking into account the reviewer’s comments. In this revision, we added our estimation of the rapid adjustment of BC to the main text. We also added comparisons of our simulations with aircraft measurements and MAC observations.

Detailed responses to individual comments and suggestions are given below.

Reviewer’s comment:

1) The manuscript lacks inclusion of the rapid adjustment associated with BC (the term semi-direct effect has previously often been used, but rapid adjustment is a broader term than semi-direct effect often associated with BC induced cloud changes. The rapid adjustment is shown to be highly important for BC in multi-model climate simulations (Stjern et al., 2017) and explains a low temperature response from BC (Baker et al., 2015; Samset et al., 2016). Since the global mean temperature change from BC in global climate models is lower than expected the rapid adjustment is essential to include to provide the full picture of BC for mitigation purposes. In several climate models the offset by the rapid adjustment of the direct radiative effect is almost 50% (Stjern et al., 2017), highlighting this as a necessary effect to include. Furthermore, the rapid adjustment of BC is shown in a single-model to vary inversely with the strength of the direct radiative effect, given a net climate impact which is less sensitive to uncertainties associated with BC (Hodnebrog et al., 2014). The effective radiative forcing (ERF) and thereby the rapid adjustment should be fairly easy to extract from the performed simulations in the manuscript by Matsui et al. Since the BC signal is rather small, expanding the number of years of simulations may be needed.

Response:

This is an interesting point, which we think we have addressed in the revised manuscript. First, we attempted to estimate the rapid adjustment associated with BC from 16-year online simulations with and without BC (Single-MS). Simulated BC rapid adjustment has a very large interannual variability (or variability/mean ratio) as shown by Figure (a) below. Therefore, the rapid adjustment estimated from these simulations has too large uncertainties (i.e. small signal-to-noise ratio) to get a climatologically useful value.

Next, we estimated the rapid adjustment of BC from the simulations with 10 times enhanced BC emission set-up ($\times 10$ BC) (Single-MS), based on the method of Stjern et al. (2017). The interannual variability of the rapid adjustment (variability/mean ratio) is much smaller in these simulations as shown by Figure (b) below, though the variability is still much greater than the interannual variability of BC DRE (Figure (b)). The fraction of the rapid adjustment to BC DRE was estimated to be $-44\pm 8\%$, $-32\pm 6\%$, and $-25\pm 10\%$ in the Small-, Base-, and Large-size simulations, respectively. The decrease in the fraction (more negative) with decreasing emission size distributions is likely due to both indirect and semidirect effects. However, this method ($\times 10$ BC) is not

available in Multiple-MS because the 10 times enhancement of BC emissions only (without changing other species) leads to unrealistic calculations of BC aging processes and mixing states (therefore unrealistic estimates of BC DRE and rapid adjustment).

Considering that the rapid adjustment of BC is much more uncertain than the BC DRE and that “ $\times 10$ BC” simulations are not available in Multiple-MS, we think adding a paragraph on the rapid adjustment to the Discussion section is the best way to describe this potentially important (but very uncertain) effect.

We added the following sentences to the Discussion section. “Recent studies have suggested that the rapid adjustment (aerosol induced cloud changes) associated with BC may be important for explaining a low temperature response from BC⁵⁹⁻⁶¹. Since the global mean temperature change from BC in global climate models is lower than expected⁶², the rapid adjustment may be an important effect required to obtain the full picture of how BC modifies climate and hence understanding future mitigation policy impacts. The estimation of the rapid adjustment of BC is much more uncertain than that of BC DRE. Here we suggest that $44\pm 8\%$, $32\pm 6\%$, and $25\pm 10\%$ of BC DRE may be offset by the rapid adjustment in the Small-, Base-, and Large-size simulations (Methods, Supplementary Fig. 9, and Supplementary Table 4), noting that the offset percentage increases with decreasing emission sizes (Supplementary Table 4). If we assume these percentages can be applied to our BC DRE estimates, the Small/Large ratios of BC effective radiative forcing (BC DRE + rapid adjustment) are reduced to 1.7 and 0.80 for Multiple-MS and Single-MS, respectively (2.3 and 1.1, respectively, for BC DRE only). These results suggest accurate estimates of the rapid adjustment combined with BC MS diversity will be important in future studies to constrain BC climate impact.” (Lines 353-367).

Figure (b) below was added to Supplementary Information as Supplementary Fig. 9. Statistics of DRE, rapid adjustment, and cloud radiative forcing from the “ $\times 10$ BC” simulations were added as Supplementary Table 4.

We also added the method for estimating the rapid adjustment to the Methods section (Lines 559-571).

Reviewer’s comment:

2) The climate effect of BC is uncertain because of many factors influencing the forcing and few constraints from observations. Anyway, we have two constraints from observations on the vertical profile of BC and mass absorption coefficient (MAC). Reference number 10 in Matsui et al. shows vertical profiles of BC compared to

observations. Since the lifetime of BC changes in model simulations, I recommend to show how the vertical profile of BC compares with observations. A standard finding in the global models is that the abundance of BC is overestimated in the upper troposphere compared to observations (Samset et al., 2014; Wang et al., 2014a) and it would be valuable to see if any of the model simulations performed in this study compares better or worse with the observations and possibly be used as a constrain. There are uncertainties in the measurements and large spatial and temporal variations of MAC, but a wide range of measurements points to a majority of MAC somewhere between 7 and 13 m²/g at 550 nm (Bond and Bergstrom, 2006; Bond et al., 2013; Zanatta et al., 2016). Are the simulated MAC values within the range from observations?

Response:

We thank the reviewer for providing us this valuable comment. We made analysis for vertical profiles and MAC values of BC.

The figure below shows comparisons of simulated BC with aircraft measurements. We added this figure to Supplementary Information as Supplementary Fig. 1. BC mass has higher sensitivity to emission size distributions over remote regions than near source regions, which is consistent with the spatial distributions of the sensitivity of BC DRE in Fig. 2 and Supplementary Figs 3 and 4. Small-size simulations tend to have a better agreement with measurements in the lower troposphere in the Northern Hemisphere, whereas Large-size simulations are closer to measurements in the upper troposphere. However, we think the differences in BC mass between simulations are not large enough to conclude which model setup is better or worse because there are large uncertainties in comparisons between measurements and model simulations, as described in Matsui and Mahowald (2017). Considering this point, we added the following sentence to the main text: “However, the overestimation of upper tropospheric BC in many global aerosol models^{30,32} is also seen in the Multiple-MS and Single-MS simulations shown here (Supplementary Fig. 1).” (Lines 106-108).

Regarding to MAC values, we can now say more strongly that Multiple-MS is more realistic than Single-MS. We added the following paragraph to the main text: “Mass absorption cross section (MAC) has large spatial and temporal variations: typically between 7 and 13 m² g⁻¹ at 550 nm in measurements and an average of ~8 m² g⁻¹ at 550 nm in AeroCom models^{1,33-37}. Global-mean MAC is 8.8 and 13.6 m² g⁻¹ in Multiple-MS and Single-MS (Base-size simulations), respectively (Supplementary Table 3), showing that MAC in Multiple-MS is within the typical observed range while MAC in Single-MS is above the typical observed range.” (Lines 130-135).

Reviewer's comment:

Line 34-36: Bond et al. (ref 1) highlight BC as the second largest positive climate forcing in terms of emissions. Note that there is a difference between describing this in terms of emissions and in abundance as the latter is implied in the manuscript. For methane the emission based forcing is much larger than the abundance based forcing. There is several published papers since 2013 indicating that Bond et al. overestimate the BC forcing. Wang et al. (2014b) shows that high resolution modelling reduce substantially the difference between model simulation of AAOD and AERONET data, indicating that AERONET stations generally is located close to BC emission sources and overemphasizing the difference between AERONET and global models.

Samset et al. (2014); Wang et al. (2014a) shows that global models generally has too long BC lifetime and overestimate the BC forcing. In addition the rapid adjustment of BC seems much more important than provided in Bond et al (2013), see point 1 above. Given the more recent findings on BC, I don't think highlighting BC as the second most important forcer is correct and think you should refrain from repeating this. Highlighting the uncertainty on the other hand is fine.

Response:

We deleted the description that BC is the second largest positive climate forcer from this sentence (Lines 30-32) as the reviewer suggested.

We also added the following sentence to the main text to cite Samset et al. (2014) and Wang et al. (2014): "Simulated BC lifetimes in both Multiple-MS (4.5-6.0d) and Single-MS (4.3-5.1d) tend to be shorter than the AeroCom multi-model average (6.5 d)³⁰, which is generally consistent with previous studies^{30,31} suggesting a too long BC lifetime in many AeroCom models." (Lines 103-106).

Reviewer's comment:

Introduction: The importance of mixing for cloud condensation nuclei (CCN) activity is mentioned several times. On line 260 it is described that aerosol-cloud interaction is included together with aerosol-radiation interaction (direct radiative effect). It would be nice to add a table separating the aerosol-cloud interaction from the aerosol-radiation interaction (as well as including the rapid adjustment as suggested in point 1 above).

Response:

We have added a table showing the DRE, rapid adjustment, and instantaneous cloud radiative effect of BC from the "×10 BC" simulations (described in the response to the first major comment) to Supplementary Information (Supplementary Table 4). However, the rapid adjustment and cloud radiative effect are much more uncertain than DRE as shown in the response to the first major comment. In addition, CAM5 tends to overestimate aerosol indirect forcing from PI to PD (about -2 W m⁻², compared with about -1 W m⁻² in IPCC AR5). Therefore, we would like to focus on BC DRE discussions in this study.

Reviewer's comment:

Line 109-110: I have problem finding the uncertainty range of 0.20 Wm⁻² from the AeroCom models in reference 1. This value is much higher than from the original AeroCom paper.

Response:

We clarified this point in the revised manuscript as follows: "Bond et al.¹ (Table 15) reported all-source BC direct radiative forcing (DRF) (difference between present-day and pre-industrial) of 0.20-0.36 W m⁻² for the AeroCom models. We assumed pre-industrial BC DRE was 20% of present-day BC DRE for these models to calculate BC DRE in the present-day based on the preindustrial (year 1850) to present-day (year 2000) fraction of BC emissions reported in Lamarque et al.⁶⁵. The maximum-minimum range is therefore 0.20 W m⁻² ((0.36-0.20)/0.8) for these models."

(Lines 481-487).

Reviewer's comment:

Line 127-132: It could be mentioned that BC burden and Eabs is related, this is shown in Boucher et al. (2016).

Response:

Since Boucher et al. (2016) discussed the importance of MAC, we cited this paper in the discussion of MAC which was added to the main text following the second major comment. (Line 132)

Reviewer's comment:

Line 333-336 & line 360-362: The uncertainty range here taken as difference between maximum and minimum values is not a standard approach. This needs to be confirmed to be similar to approaches that are more standard.

Response:

Model intercomparison studies usually use 5-95% uncertainty ranges. Estimation of a 5-95% uncertainty range is useful when the number of models and/or simulations allow it. However, since this study is a single-model study and each simulation requires large amounts of computational cost, it is not practical to estimate the 5-95% uncertainty range from the limited number of simulations in this study. In model intercomparison studies (e.g., Myher et al. 2013), "range" is sometime used as a maximum-minimum difference. Considering this point, we changed "uncertainty range" to "range" or "sensitivity" in the revised manuscript.

We added this discussion to the Method section (Lines 490-495).

Reviewer's comment:

Line 338-342: This is very unclear. Do you describe anthropogenic BC emission prior to 1750 (or 1850)?

Response:

We clarified that the PI/PD fraction is based on an emission dataset (Lamarque et al. 2010): "We assumed pre-industrial BC DRE was 20% of present-day BC DRE for these models to calculate BC DRE in the present-day based on the preindustrial (year 1850) to present-day (year 2000) fraction of BC emissions reported in Lamarque et al.⁶⁵." (Lines 483-486). PI simulations are not conducted in this study.

Reviewer's comment:

Line 348-360: Given the concern that Bond et al (ref 1) overestimate the forcing of BC (including the DRE), I recommend to not scale your results to the Bond et al estimate.

Response:

We deleted comparisons with the range in Bond et al. BC assessment based on this reviewer's comment. This deletion does not change our main conclusions because we mainly use Figure 3 (comparisons with the AeroCom models) for comparisons with

the ranges by other aerosol models.

Reviewer's comment:

Line 416-418: The values quoted from (Myhre et al., 2013) (ref 3) is DRE (RFari in IPCC AR5 terminology) and not ERF as given in the text.

Response:

We changed ERF to DRF in this sentence (Line 551) as the reviewer suggested.

References:

Bond, T. C. *et al.* Bounding the role of black carbon in the climate system: A scientific assessment. *J. Geophys. Res. Atmos.* **118**, 5380-5552 (2013).

Boucher, O. *et al.* Jury is still out on the radiative forcing by black carbon. *Proc. Natl. Acad. Sci. USA* **113**, E5092-E5093 (2016).

Lamarque, J.-F. *et al.* Historical (1850-2000) gridded anthropogenic and biomass burning emissions of reactive gases and aerosols: methodology and application. *Atmos. Chem. Phys.* **10**, 7017-7039 (2010).

Matsui, H. & Mahowald, N. Development of a global aerosol model using a two-dimensional sectional method: 2. Evaluation and sensitivity simulations. *J. Adv. Model. Earth Syst.* **9**, 1887-1920 (2017).

Myhre, G. *et al.* Radiative forcing of the direct aerosol effect from AeroCom Phase II simulations. *Atmos. Chem. Phys.* **13**, 1853-1877 (2013).

Samset, B. H. *et al.* Modelled black carbon radiative forcing and atmospheric lifetime in AeroCom Phase II constrained by aircraft observations. *Atmos. Chem. Phys.* **14**, 12465-12477 (2014).

Stjern, C. W. *et al.* Rapid adjustments cause weak surface temperature response to increased black carbon concentrations. *J. Geophys. Res. Atmos.* **122**, 11462-11481 (2017).

Wang, Q. *et al.* Global budget and radiative forcing of black carbon aerosol: Constrains from pole-to-pole (HIPPO) observations across the Pacific. *J. Geophys. Res. Atmos.* **119**, 195-206 (2014).

Reviewers' Comments:

Reviewer #1 (Remarks to the Author):

This manuscript presents the results from a novel analysis which identifies that variations in the size distribution of emitted black carbon (BC) introduces a major uncertainty in the magnitude of the direct radiative effects from black carbon (BC). Furthermore, the paper explains how conventional aerosol models, which do not represent variations in BC mixing state, are under-representing this strong influence on the efficacy of BC radiative effects.

The authors reach these findings by applying the model described and evaluated in Matsui (2017) and Matsui and Mahowald (2017), to realistically resolve the different particle morphologies that exist and evolve following emission from different sources and subsequent chemical and microphysical transformations which occur during transport.

The analysis in the paper decomposes this "BC emissions-size-sensitivity effect" into its three main driving factors: BC lifetime, mass-absorption efficiency and core-shell absorption enhancement. In so-doing, the analysis then is able to illustrate why resolving the BC mixing state causes such an elevated sensitivity to variations in emissions particle size. In all three of these factors, resolving the internal mixing variations for the BC causes a stronger sensitivity to variations in emitted particle size.

In my initial review from December 2017, I found the analysis sufficiently novel and important as to be publishable in Nature Communications, but I raised a concern that in several parts of the manuscript, the writing could be improved to better explain the findings. In particular I asked for Figure 2 (Figure 4 in the revised manuscript) to be improved to better explain the driving factors behind the high DRE sensitivity with the multiple MS representation, and for the equations to be improved and clarified.

As well as a list of specific minor revisions, I also identified two major issues that I asked the authors to clarify in their response and make clearer in the revised manuscript. Firstly I queried the way the paper was comparing to the scaled multi-model range from the Bond et al. (2013) BC assessment. Secondly I explained the issue that even global aerosol models with simplified BC treatments would resolve aerosol optical properties representative of fresh and aged mixed BC-POM particles (even if they only resolved BC mass in

the model).

In my review I clarified that, despite major revisions being required, I thought it likely the authors would be able to address these, and thereby revise the manuscript to ensure it communicates the important findings more effectively.

I have read carefully the responses of the authors to the comments I made, and can see that they have greatly improved the manuscript, addressing almost all the comments I made, including the two major issues. The scaled-range from the Bond et al. (2013) BC assessment (Figure 1f) is now removed, and the authors have added new results from offline radiative calculations which now address this issue. Important clarifying/qualifying text that much better sets their results in context has also been added.

The authors are to be congratulated for improving the text and Figures to now be much clearer and carefully worded. Having read through all the changes, I am now content to approve the paper to be published in Nature Communications with only very minor typographical changes needed.

1) Page 12, line 248 -- insert "being" before "enhanced by aerosol water" and change "aerosol water" to "the presence of water in the particle".

2) Page 20, line 440 -- replace "Their ranges of emissions particle sizes correspond.." with "This uncertainty range for these emissions particle sizes corresponds..."

3) Page 20, line 443-445 -- Re-word this sentence, replacing "The range for..." with "The emissions particle size range for...", replace "are slightly larger the AeroCom models (30-150nm)" with "is at slightly larger sizes compared to the diversity among AeroCom models (30-150nm)", and also reword the last section from "but a recently updated study by that group suggests carbonaceous biomass burning emissions could have maximum..." with "and a more recent aerosol forcing uncertainty study (by the same group) suggests emitted carbonaceous particles from biomass burning may be as large as..."

Reviewer #3 (Remarks to the Author):

Review of manuscript by Matsui et al., entitled 'Impact of particle size and mixing state diversity on estimates of black carbon mitigation'. In general, I think the authors have made a nice job in

responding to my first round of comments.

My only main concern with the current version of the manuscript is associated with the single MS. In the new version of Fig 4 (earlier Fig 2), it is clear that this BC scheme is simpler than used in most global aerosol models. Most of the simpler aerosol models have one hydrophobic and one hydrophilic BC component. I think using a BC scheme with one hydrophobic and one hydrophilic BC would influence both the lifetime and MACcore calculations. I would therefore encourage the authors to investigate the range in the forcing from such a BC scheme. Sorry, that I didn't comment on this point in the first round!

Since it seems as the authors would like to keep 'mitigation' in the title, I think you should strengthened the concern of BC for climate mitigation purposes based on your findings on emission sizes, but also due to uncertainties in rapid adjustments and aerosol-cloud interactions from BC. You discuss this in the last section, but it could be strengthened and also included in the abstract.

Line 124-125: It is unclear which other aerosol species you refer to here. Mineral dust and sea salt are obviously sensitive to emission sizes and other main aerosol species are secondary aerosols (sulphate, nitrate, SOA).

Line 208: Are there no cases where BC is negligible so you have BC-free particles in single-MS?

Line 297: The very strong total DRE efficiency in single MS for BB emissions are not in line with other global aerosol models. Do you have sufficient OM included in your model?

Line 302: Here you can mention that BB emissions have other climate effects than the direct aerosol effect such as the rapid adjustments and aerosol-cloud interactions.

I support the authors on using AAOD550, with 550 in lowercase.

Response to reviewer #1

NCOMMS-17-30621A: "Impact of particle size and mixing state diversity on estimates of black carbon mitigation" by H. Matsui et al.

We thank the reviewer very much for reading the paper carefully and giving us valuable comments. We revised the paper by taking into account the reviewer's comments. Detailed responses to individual comments and suggestions are given below.

Reviewer's comment:

Page 12, line 248 -- insert "being" before "enhanced by aerosol water" and change "aerosol water" to "the presence of water in the particle".

Response:

We revised the sentence as the reviewer suggested (Lines 262-263).

Reviewer's comment:

Page 20, line 440 -- replace "Their ranges of emissions particle sizes correspond.." with "This uncertainty range for these emissions particle sizes corresponds..."

Response:

We revised the sentence as the reviewer suggested (Lines 460-461).

Reviewer's comment:

Page 20, line 443-445 -- Re-word this sentence, replacing "The range for..." with "The emissions particle size range for...", replace "are slightly larger the AeroCom models (30-150nm)" with "is at slightly larger sizes compared to the diversity among AeroCom models (30-150nm)", and also reword the last section from "but a recently updated study by that group suggests carbonaceous biomass burning emissions could have maximum...." with "and a more recent aerosol forcing uncertainty study (by the same group) suggests emitted carbonaceous particles from biomass burning may be as large as...".

Response:

We revised the sentence as the reviewer suggested (Lines 464-467).

Response to reviewer #3

NCOMMS-17-30621A: “Impact of particle size and mixing state diversity on estimates of black carbon mitigation” by H. Matsui et al.

We thank the reviewer very much for reading the paper carefully and giving us valuable comments. We revised the paper by taking into account the reviewer’s comments. Detailed responses to individual comments and suggestions are given below.

Reviewer’s comment:

My only main concern with the current version of the manuscript is associated with the single MS. In the new version of Fig 4 (earlier Fig 2), it is clear that this BC scheme is simpler than used in most global aerosol models. Most of the simpler aerosol models have one hydrophobic and one hydrophilic BC component. I think using a BC scheme with one hydrophobic and one hydrophilic BC would influence both the lifetime and MACcore calculations. I would therefore encourage the authors to investigate the range in the forcing from such a BC scheme. Sorry, that I didn’t comment on this point in the first round!

Response:

We would like to thank the reviewer for providing us this important comment. We added simulations using a 2 MS representation (Double-MS), which resolves thinly-coated BC (BC mass fraction of 0.90-1.00) and the other particles, to the manuscript (Figs 1 and 3, Supplementary Figs 2, 3, 5, and 11, and Supplementary Table 1). We also added the following paragraph to the main text: “The simulations assuming 2 MS categories (Double-MS; thinly-coated BC and others) (Supplementary Table 1) have a slightly higher sensitivity of BC lifetime, AAOD₅₅₀, and DRE to emission size distributions than the Single-MS simulations (green lines in Figs 1 and 3) because Double-MS resolves thinly-coated BC which has a longer lifetime and lower MAC than the internally-mixed BC represented in Single-MS. However, as shown in Figs 1 and 3, the sensitivity of Double-MS and Single-MS is similar for all BC properties. This result suggests that most global aerosol models, which use a BC representation similar to Double-MS or Single-MS, currently underestimate the sensitivity of BC properties with respect to emission size distributions.” (Lines 175-183).

Reviewer’s comment:

Since it seems as the authors would like to keep ‘mitigation’ in the title, I think you should strengthened the concern of BC for climate mitigation purposes based on your findings on emission sizes, but also due to uncertainties in rapid adjustments and aerosol-cloud interactions from BC. You discuss this in the last section, but it could be strengthened and also included in the abstract.

Response:

We changed the title of this manuscript and removed “mitigation” from it. We added the importance of rapid adjustments and aerosol-cloud interactions to the manuscript as follows: “In addition to reducing other uncertainties of BC such as

emission mass flux, aerosol-cloud interactions, and rapid adjustments, a more detailed representation of aerosol size distributions and MS is therefore required when evaluating the effectiveness of BC mitigation in order to make effective climate change policy.” (Lines 389-393).

Reviewer’s comment:

Line 124-125: It is unclear which other aerosol species you refer to here. Mineral dust and sea salt are obviously sensitive to emission sizes and other main aerosol species are secondary aerosols (sulphate, nitrate, SOA).

Response:

We clarified that only emission sizes for fossil fuel, biofuel, and biomass burning sources are changed in this study as follows: “Only the treatment of emission size distributions for anthropogenic (AN; fossil fuel and biofuel) and biomass burning (BB) sources is different in these 10 simulations.” (Lines 85-87 and Lines 453-454). We also clarified that other aerosol species are sulfate, nitrate, dust, and sea salt (Line 126).

Reviewer’s comment:

Line 208: Are there no cases where BC is negligible so you have BC-free particles in single-MS?

Response:

We revised the text as follows: “BC-free particles are not treated (0%) in Single-MS unless BC is negligible” (Lines 223-224).

Reviewer’s comment:

Line 297: The very strong total DRE efficiency in single MS for BB emissions are not in line with other global aerosol models. Do you have sufficient OM included in your model?

Response:

In our simulations, OA emissions from biomass burning sources are 33.4 Tg y^{-1} and are the same as the emissions in Liu et al. (2012). The positive DRE efficiency in Single-MS may be because MAC is overestimated in Single-MS.

Reviewer’s comment:

Line 302: Here you can mention that BB emissions have other climate effects than the direct aerosol effect such as the rapid adjustments and aerosol-cloud interactions.

Response:

Considering this reviewer’s comment, we revised this sentence as follows: “Though BB emissions also change aerosol-cloud interactions and rapid adjustments, this result suggests that the enhancement of BB emissions in the future will have an overall cooling impact on climate and hence does not hinder the effect of BC mitigation from industrial sources.” (Lines 312-315).

Reference:

Liu, X. *et al.* Toward a minimal representation of aerosols in climate models: description and evaluation in the Community Atmospheric Model CAM5. *Geosci. Model Dev.* **5**, 709-739 (2012).